# scDREAMER for atlas-level integration of single-cell datasets using deep generative model paired with adversarial classifier

Ajita Shree [1,4], Musale Krushna Pavan [1,4] & Hamim Zafar [1,2,3] ✉

Integration of heterogeneous single-cell sequencing datasets generated across multiple tissue locations, time, and conditions is essential for a comprehensive understanding of the cellular states and expression programs underlying complex biological systems. Here, we present scDREAMER (https://github.com/Zafar-Lab/scDREAMER), a data-integration framework that employs deep generative models and adversarial training for both unsupervised and supervised (scDREAMER-Sup) integration of multiple batches. Using six real benchmarking datasets, we demonstrate that scDREAMER can overcome critical challenges including skewed cell type distribution among batches, nested batch-effects, large number of batches and conservation of development trajectory across batches. Our experiments also show that scDREAMER and scDREAMER-Sup outperform state-of-the-art unsupervised and supervised integration methods respectively in batch-correction and conservation of biological variation. Using a 1 million cells dataset, we demonstrate that scDREAMER is scalable and can perform atlas-level cross-species (e.g., human and mouse) integration while being faster than other deep-learning-based methods.

The exploration of cellular heterogeneity and developmental trajectories in different tissue systems has been revolutionized by the rapid advances of single-cell RNA-sequencing technologies[1–3]. This rapid development coupled with large-scale collaborative initiatives such as the Human Cell Atlas (HCA) and Human BioMolecular Atlas Program (HuBMAP)[4,5] have increased the complexity of single-cell datasets which can include samples contributed by different laboratories[6], generated across tissue locations, time and conditions[7,8]. Since single-cell datasets generated from similar biological contexts but different experimental conditions can share cellular features[9], integration of information from heterogeneous data sources can facilitate the discovery of major and rare cell types, improve the reconstruction of developmental trajectories[10] and lead to more reliable investigation of complex biological systems. However, the intrinsic differences in measured gene expression among experimental settings contributed

by factors such as sequencing protocols, library preparation, sample donors, tissue of origin, sampling time and condition inevitably create complex, nested batch effects that can diminish the value of data integration by confounding biological signals[11]. Thus the development of computational data integration methods that can reliably eliminate the complex batch effects without undermining biological variations is a major challenge in scRNA-seq analysis[12].

Existing methods for the integration of scRNA-seq datasets can be broadly classified into two groups. The first group of methods employs cell type annotations for supervised cross-batch learning[13,14] while removing batch effects. The requirement for cell type annotations limits their applications as novel cell types cannot be captured. In comparison, unsupervised data integration methods[15–18] that do not require cell type annotations are more widely used. Some of these methods identify the batch-specific gene factors from the gene expression profiles[19].

[1]Department of Computer Science and Engineering, Indian Institute of Technology Kanpur, Kanpur, India. [2]Department of Biological Sciences and Bioengineering, Indian Institute of Technology Kanpur, Kanpur, India. [3]Mehta Family Centre for Engineering in Medicine, Indian Institute of Technology Kanpur, Kanpur, India. [4]These authors contributed equally: Ajita Shree, Musale Krushna Pavan. ✉e-mail: hamim@iitk.ac.in

Methods such as BBKNN, Scanorama and Seurat v3 employ global mutual nearest neighbors (MNNs), i.e., paired cells between multiple batches for batch correction of the neighborhood using reduced-dimension cellular spaces. Harmony[15] achieves better batch mixing by applying a novel local correction. These MNN-based methods consume a large memory and low-quality MNNs can make it difficult to simultaneously identify dataset-specific cell types and the cell types that are shared by multiple datasets. Methods such as scVI[20] and DESC[21] that employ deep variational autoencoders for learning cellular embeddings from scRNA-seq can also integrate data from multiple batches. However, the traditional autoencoder models are challenged due to lower fidelity in reproducing the batch-corrected expression profiles[22]. Another method, iMAP[22] combines generative adversarial networks (GANs) with autoencoder for learning batch-ignorant cellular representations, however its dependence on MNN pairs to train the GAN can lead to sub-optimal integration due to low-quality MNN and unstable training of GAN[23]. A recent benchmarking study[24] showed that methods such as Harmony and Seurat perform well for simple integration tasks but poorly for complex integration tasks and vice versa for Scanorama and scVI and a consistent tradeoff exists between batch-correction and preservation of biological variations.

To overcome the existing challenges, here we present a deep learning-based data integration framework, called scDREAMER (single-cell Deep geneRativE integrAtion Model with advErsarial classifieR) that performs the integration of multiple batches in unsupervised (no cell type annotations required), semi-supervised and supervised (available cell type annotation are utilized) manner. The unsupervised version of scDREAMER employs an adversarial variational autoencoder and a batch classifier (a multi-layer neural network) which are trained adversarially for learning batch-invariant lower-dimensional cellular embeddings. The supervised version, scDREAMER-Sup, employs an additional variational autoencoder and a cell type classifier (another feed-forward neural network) to utilize available cell type annotations for a semi-supervised or supervised inference of cellular latent spaces. Using multiple real datasets consisting of up to 1 million cells and 147 batches (Supplementary Table 1), we demonstrate that scDREAMER is able to overcome a variety of integration challenges including the presence of skewed cell types among batches (pancreas integration), nested batch effects (lung integration), large number of batches (heart atlas and macaque retina integration) and conservation of development trajectory across different batches (human immune integration). Using these integration tasks, we further show that scDREAMER achieves better performance in batch-correction and conservation of biological variation against that of the state-of-the-art unsupervised methods. The supervised version of scDREAMER, scDREAMER-Sup improves upon scDREAMER in terms of conservation of biological variation while retaining superior performance in batch-correction and outperforms other state-of-the-art supervised and unsupervised integration methods. For a challenging heart atlas dataset, which several methods failed to integrate due to a large number of batches and complex nested batch effects, scDREAMER and scDREAMER-Sup outperform the other methods by a large margin. Our experiments also show that scDREAMER and scDREAMER-Sup can reliably identify rare cell types. Using semi-supervised integration settings, we also demonstrate scDREAMER-Sup's superiority over other methods in predicting the cell type labels for the cells missing annotations. Finally, using a 1 million cells dataset, we demonstrate that scDREAMER is highly scalable, can perform atlas-level data integration across different species, and achieves runtime advantage over some other deep-learning-based integration methods.

## Results
### Overview of scDREAMER
Figure 1 shows the overview of both the unsupervised and supervised models of scDREAMER. The unsupervised model (we will refer it to as

scDREAMER) employs an adversarial variational autoencoder for learning the lower-dimensional representation of cells from the high-dimensional scRNA-seq data and a neural network classifier (also called a batch classifier) for the removal of batch effects. scDREAMER models the scRNA-seq data as a nonlinear function of a lower-dimensional cell-state embedding and the batch information that encodes the variation in data generation. The adversarial variational autoencoder of scDREAMER consists of three multi-layer neural networks: an encoder $E$ that maps the high-dimensional expression data ($x_i$) and batch information ($s_i$) of a cell $i$ to a lower-dimensional embedding $z_i$, a decoder $D$, which reconstructs the expression profile of the cell from $z_i$ and $s_i$, and a discriminator $\mathcal{D}$ that aims to distinguish the original expression profile $x_i$ and the expression profile reconstructed ($\bar{x}_i$) by the decoder. The adversarial variational autoencoder network of scDREAMER is trained using two loss functions: evidence lower bound (ELBO) is used for training the encoder and decoder networks, whereas Bhattacharyya loss is used for adversarial training of discriminator and autoencoder parameters. scDREAMER further incorporates a batch classifier $\mathcal{B}$ (a multi-layer neural network) that takes as input the lower-dimensional embedding $z_i$ learned by the encoder and tries to predict the batch information for cell $i$. The batch classifier and the encoder are adversarially trained using a cross-entropy loss where the encoder tries to maximize it with an aim to generate the embeddings such that the classifier is not able to differentiate between batches and the batch classifier tries to minimize it by distinguishing the embeddings of the cells coming from different batches and hence achieving better mixing of the batches (see "Methods" section for details).

We further extended the scDREAMER model to utilize available cell type annotations (Fig. 1) for a guided inference of cellular latent space $z_i$. The extended model, scDREAMER-Sup assumes an informative prior on $z_i$ conditioned on cell type label $c_i$ and another Gaussian latent variable $y_i$ that accounts for within cell type variability. For learning the hierarchically structured $z_i$, scDREAMER-Sup employs an additional variational autoencoder consisting of an encoder $E_y$ that learns $y_i$ from $z_i$ and $c_i$ and a decoder $D_y$ that reconstructs $z_i$ from $y_i$ and $c_i$ such that a more informative prior can be used for $z_i$ in the adversarial variational autoencoder. The two variational autoencoders are jointly trained using an updated ELBO function (see "Methods" for details). scDREAMER-Sup further employs a feed-forward neural network (also called cell type classifier), $\mathcal{C}$, for learning $c_i$ from $z_i$ which is trained using a cross-entropy loss using the available cell type annotations. For cells without cell type annotations, scDREAMER-Sup learns the missing cell type labels using $\mathcal{C}$. $\mathcal{B}$ and $\mathcal{D}$ are trained in the same way as in scDREAMER.

### scDREAMER integrates pancreatic islet data generated using different sequencing protocols
We first tested scDREAMER's ability to perform integration and batch correction across different sequencing protocols using a human pancreas dataset consisting of 16,382 cells (Supplementary Fig. 1a). The dataset consisted of nine sub-datasets generated using distinct sequencing protocols including CEL-seq, CEL-seq2, Fluidigm C1, SMART-seq2, and inDrop. The dataset harbored 14 pancreatic cell types including acinar cells, activated and quiescent stellate cells, alpha cells, beta cells, delta cells, ductal cells, endothelial cells, epsilon cells, gamma cells, macrophages, mast cells, Schwann cells and T cells. scDREAMER's integration performance was compared against that of nine other integration methods. scDREAMER was able to almost perfectly separate all the cell types and mix the shared cell types across different protocols (Fig. 2a, b). In comparison, integration by scVI, BBKNN, Scanorama, INSCT, and iMAP led to restricted mixing of the different batches (Supplementary Fig. 1b–j). While Harmony, Seurat, scDML and LIGER were also able to mix the batches well, Harmony was not able to clearly distinguish activated and quiescent stellate cells,

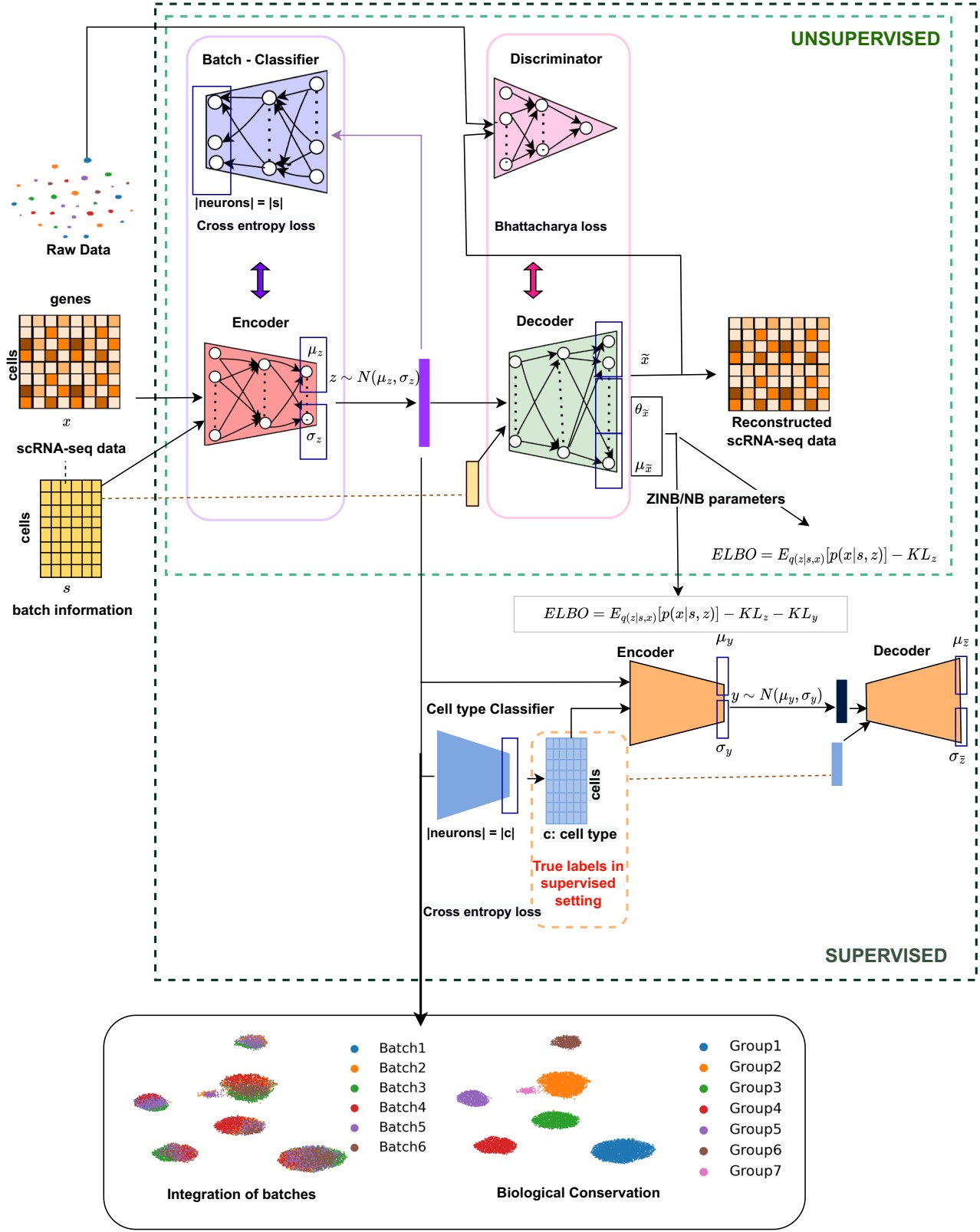

**Fig. 1 | Overview of scDREAMER and scDREAMER-Sup.** scDREAMER consists of an adversarial variational autoencoder and a batch classifier. The adversarial variational autoencoder comprises of three networks: an encoder, a decoder, and a discriminator, and these networks are trained using ELBO and Bhattacharya loss functions. The batch classifier is adversarially trained along with the encoder using a cross-entropy loss. scDREAMER learns latent cellular embeddings such that the cells from different batches are well-mixed and different cell types are separated

leading to the conservation of biological variations. scDREAMER-Sup consists of an additional variational autoencoder and a cell-type classifier in addition to the components in scDREAMER. The hierarchical variational autoencoder is trained using an ELBO loss. The cell type classifier is trained using a cross-entropy loss. scDREAMER-Sup learns latent cellular embeddings such that the cells from different batches are well-mixed with improved conservation of biological variations.

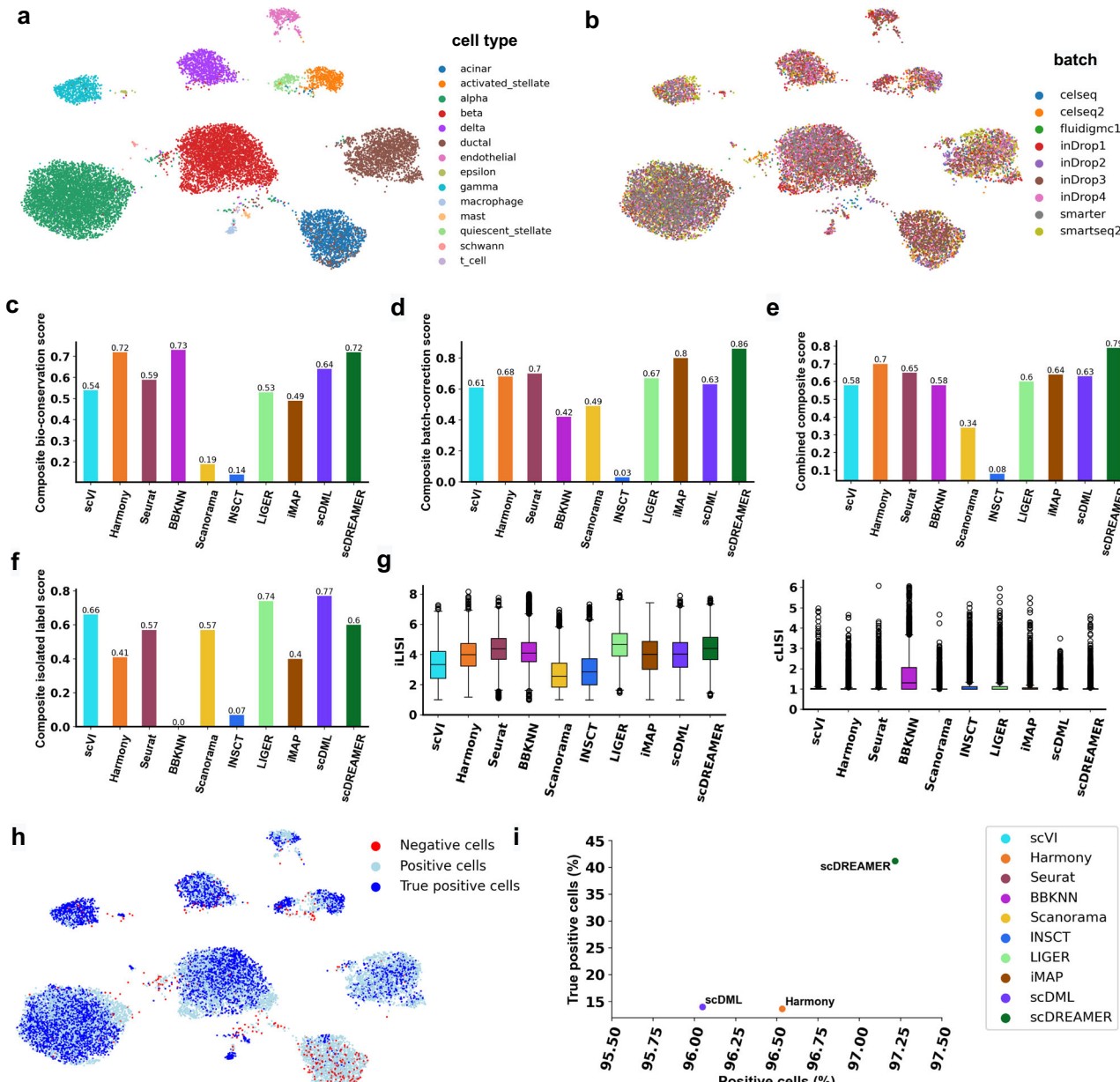

**Fig. 2 | Integration of pancreatic islet data. a** Visualization of scDREAMER's latent space embeddings after integration of pancreatic islet dataset. Different colors denote different pancreatic cell types. **b** Visualization of scDREAMER's latent space embeddings, cells are colored based on batch information. Comparison of **c** composite bio-conservation score, **d** composite batch-correction score and **e** combined composite score metrics between scVI, Harmony, Seurat, BBKNN, Scanorama, INSCT, LIGER, iMAP, scDML and scDREAMER. **f** Comparison of composite isolated label scores to assess how well rare cell types are identified. **g** Comparison of iLISI and cLISI values. Each box-and-whisker plot summarizes LISI values (n = 8208 cells, ~50% of the cells in the dataset as suggested in ref. 24), the box denotes the interquartile range (IQR, the range between the 25th and 75th percentile) with the median value, whiskers indicate the maximum and minimum value within 1.5 times the IQR, outliers are denoted by black circles. **h** Qualitative assessment of batch-mixing by visualization of scDREAMER's latent space embeddings, cells are colored based on three categories—positive, negative and true positive. **i** Quantitative assessment of batch-mixing of scDREAMER against LIGER and Harmony based on the percentage of positive vs true positive cells. Source data are provided as a Source Data file.

Seurat improperly mixed some alpha cells with beta and ductal cells, scDML fragmented alpha and beta cells into multiple clusters and mixed some alpha cells with ductal and acinar cells, and LIGER mixed some alpha, beta, and gamma cells.

Next, we quantitatively compared scDREAMER's performance against that of the other methods based on four composite accuracy scores. Composite bio-conservation score measures the accuracy of a method in preserving biological variance after integration and considers global clustering accuracy (normalized mutual information (NMI) and Adjusted Rand Index (ARI)) and relative distances between clusters (cell type average silhouette width (ASW)). The accuracy of batch effect removal was measured using a composite batch-correction score which considers four different metrics including the k-nearest-neighbor batch effect test (kBET), ASW across batches, k-nearest-neighbor (kNN) graph connectivity and batch removal using PCA regression. The combined composite score computes the average of composite bio-conservation and composite batch-correction scores. We also used a composite isolated label score that evaluates the ability of a method to capture rare cell identities based on f1 score and silhouette coefficient in identifying the rare cells.

scDREAMER consistently outperformed all other methods by achieving the highest combined composite score (-12.9% improvement over the second-best method) which was driven by scDREAMER's superior performance in both conservation of biological variance and batch correction (Fig. 2c–e). While scVI and Harmony performed similar to scDREAMER in terms of NMI and ARI metrics, scDREAMER, scDML and Harmony performed better in terms of cell type ASW (Supplementary Fig. 3a). scDREAMER achieved the best composite batch-correction score because of its superior performance in terms of multiple batch-correction metrics (Supplementary Fig. 3b). scDREAMER also performed well in capturing the rare cell identities as it achieved the second-best isolated f1 score (Fig. 2f, Supplementary Fig. 2c). While Harmony was the second-best method in terms of the combined composite score, it performed poorly in terms of composite isolated label score indicating its inability to capture the rare cell identities correctly. We also measured the local inverse Simpson's Index scores (iLISI for batch mixing and cLISI for cell-type separation) at the single-cell level and based on these metrics, scDREAMER's performance was comparable to that of Harmony and Seurat and better than all other methods (Fig. 2g). We further adopted another single-cell level evaluation measure which computes the proportion of "positive" (cells connected to other cells only from the same cell type) and "true positive" (fraction of positive cells whose local and global batch distributions are congruous) cells after integration and used these summary metrics to compare scDREAMER's performance against that of Harmony (second-best method in terms of combined composite score) and scDML (best method in terms of isolated label score). Again, scDREAMER outperformed both Harmony and scDML based on the proportion of positive and true positive cells (Fig. 2h, i).

To determine scDREAMER's ability to capture novel cell types, we performed two experiments where a specific cell type (alpha and delta cells respectively) was held out during training and the trained network was later used to obtain their embeddings. For both the held-out cell types, scDREAMER captured them very well in a separate cluster (also indicated by high values of composite bio-conservation and batch correction score for the held-out cell type) (Supplementary Fig. 3a–c) and as scDREAMER's network weights got updated after including the held-out cell type in training for certain epochs, their representation further improved as indicated by an improvement in the combined composite score (Supplementary Fig. 3c).

## scDREAMER integrates lung cells obtained from different human donors

Next, we applied scDREAMER for the integration of lung atlas data consisting of 32,472 cells from lung transplant and biopsy samples from 16 donors sequenced using 10X and drop-seq (Supplementary Fig. 4a). The integration of this dataset poses several challenges including inter-donor variability, protocol-specific batch effects (10X samples A1-A6, 1-6; drop-seq samples B1-B4), and variability across sampling type and tissue locations. Specifically, cell type composition varied between the transplant samples (1-6, B1-B4) that are obtained from lung parenchyma and the biopsy samples (A1-A6) obtained from lung airways. Two basal cell types, ciliated and secretory cells were majorly present in the biopsy samples but the transplant samples harbored them only as minor populations or in some cases the cell types were absent. The dataset also contained rare cell types present across a few donors (ionocytes). Furthermore, the endothelial and secretory cells should vary across biopsy and transplant samples due to transcriptome being affected by tissue location.

scDREAMER was able to successfully integrate the batches across sequencing protocols overcoming donor-level variation (Fig. 3a, b, Supplementary Fig. 5a, b). Only scDREAMER, scVI, and iMAP were clearly able to identify rare ionocytes as a separate cluster, whereas the other methods mixed them with other cell types (Fig. 3a, Supplementary Fig. 4b–j). scDREAMER was able to identify all biopsy-specific

cell types and also preserved basal cell subtypes. The secretory cells from the biopsy and transplant samples were separated by scDREAMER (Fig. 3a). The lymphatic and endothelial cells were incorrectly merged by Harmony, Seurat and INSCT whereas Scanorama, scVI, LIGER and scDREAMER were able to identify them as separate clusters (Supplementary Fig. 4b–j). The endothelial cells were separated into three clusters by Scanorama, two of which corresponded to transplant samples indicating incomplete integration of these samples. Integration of iMAP was poor as it divided multiple cell types (e.g., macrophages, endothelial cells, etc.) into different batch-specific clusters. Integration by scDML also led to the merging of some cell types (macrophages, basal 1 and 2, dendritic cells) and fragmentation of the same cell type (T/NK cells, Type 2 cells) into multiple clusters (Supplementary Fig. 4j). scDREAMER was able to preserve the spatial variation of endothelial cells at a higher resolution.

scDREAMER performed superior to all other methods in terms of both bio-conservation and batch correction achieving the highest combined composite score (Fig. 3c–e). scDREAMER outperformed all the methods in terms of all three bio-conservation metrics (Supplementary Fig. 6a) and also performed well in terms of different batch correction metrics (Supplementary Fig. 6b). In capturing rare cell identities, scDREAMER performed comparably with the top-performing methods (Fig. 3f, Supplementary Fig. 6c) achieving the highest isolated f1 score. In terms of LISI metrics, scDREAMER performed the best in terms of cLISI and comparably to scVI and Harmony in terms of iLISI metrics outperforming all other methods (Fig. 3g). We further compared scDREAMER's performance against that of scVI and Harmony (second and third-best methods based on combined composite score respectively) by measuring the proportion of positive and true positive cells. Due to the close spacing of dendritic cells and Neutrophil subtypes in the embedding, we observed the presence of a large number of negative cells for these cell types (same observation for the embedding of other methods). While scVI had the highest proportion of positive cells, scDREAMER outperformed both methods based on the proportion of true positive cells (Fig. 3h, i) without sacrificing much in terms of positive proportion.

## scDREAMER integrates human immune cells from peripheral blood and bone marrow of different donors

We next evaluated scDREAMER's integration and batch correction performance for integrating 33,506 human immune cells obtained in ten batches corresponding to different donors and the cells were sampled from bone marrow and peripheral blood (PBMCs) and the sequencing was performed using two protocols (10X and smart-seq2) (Supplementary Fig. 7a). Out of 33,506 cells, 9581 cells were sampled from bone marrow and comprised of three batches (Oetjen et al.[25]), and the rest 23,985 bone marrow cells comprised of seven batches (10X Genomics[26], Freytag[27], Sun et al.[28] batches and Villani et al.[29]). While the cells in ref. 29 were sequenced using smart-seq2, all other batches were sequenced using 10X Genomics protocol. This integration task poses several challenges including donor-level variability, sequencing protocol-specific batch effects, and cell types spanning multiple tissues. In addition, the dataset also contained cell subtypes that are difficult to distinguish because of transcriptional similarity (e.g., CD8[+] and CD4[+] T cells; CD14[+] and CD16[+] monocytes) and some tissue-specific cell types (e.g., monocyte progenitors, erythroid progenitors, erythrocytes and CD10[+] B cells were only present in bone marrow) that need to be identified as separate clusters. Finally, the dataset also harbored the developmental trajectory of erythrocytes from hematopoietic stem and progenitor cells (HSPCs) via megakaryocyte and erythroid progenitors and the conservation of this trajectory across batches is an important aspect of this integration task.

scDREAMER was able to resolve the inter-sample and inter-platform batch effects while integrating the human immune dataset as indicated by disjoint well-mixed cell type clusters (Fig. 4a, b). scDREAMER was

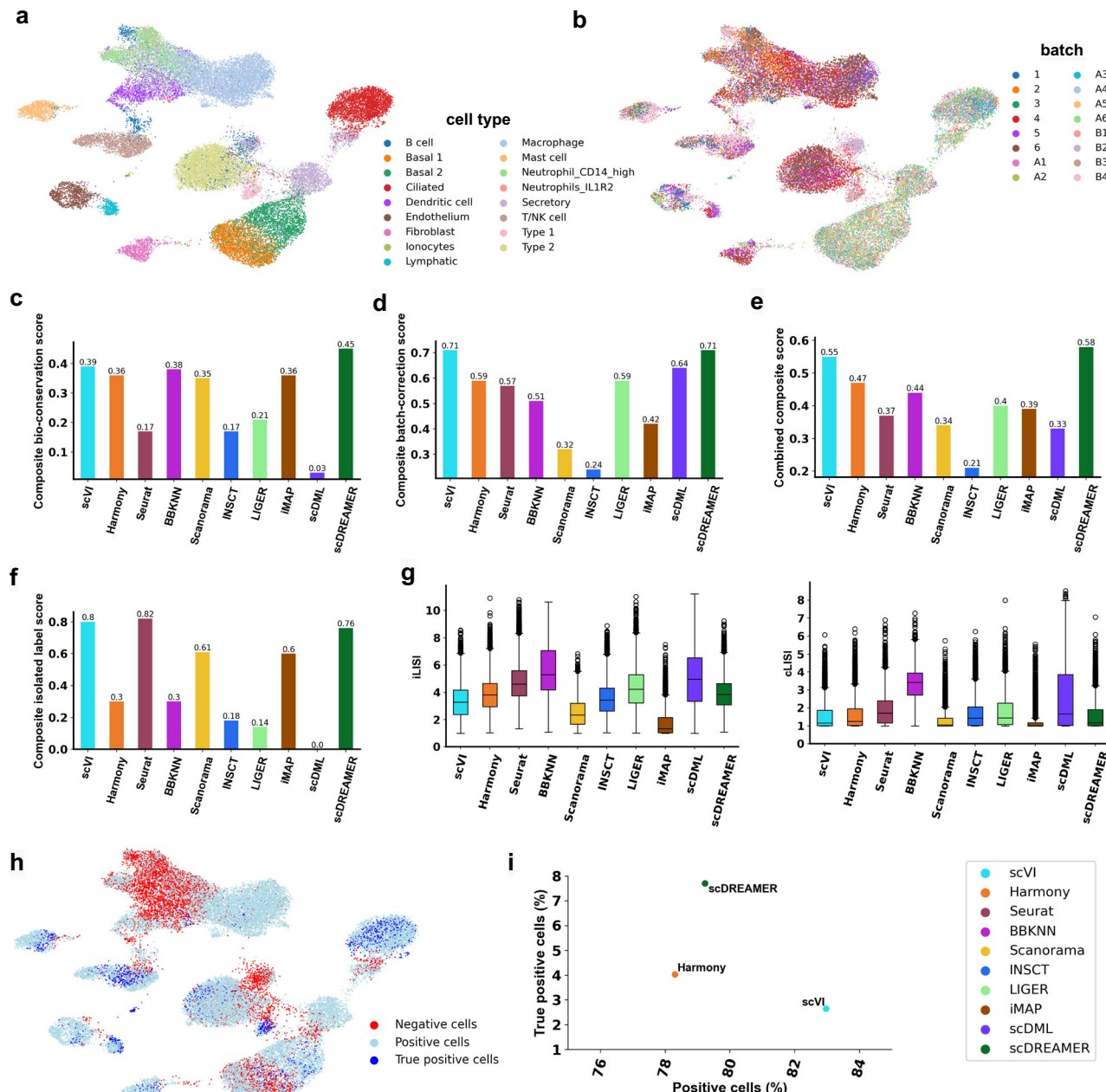

**Fig. 3 | Integration of lung atlas data. a** Visualization of scDREAMER's latent space embeddings after integration of lung atlas dataset. Different colors denote different lung cell types. **b** Visualization of scDREAMER's latent space embeddings, cells are colored based on the batch information. Comparison of **c** composite bio-conservation score, **d** composite batch-correction score and **e** combined composite score metrics between scVI, Harmony, Seurat, BBKNN, Scanorama, INSCT, LIGER, iMAP, scDML and scDREAMER for the integration of lung atlas data. **f** Comparison of composite isolated label scores to assess how well rare cell types are identified. **g** Comparison of iLISI and cLISI values. Each box-and-whisker plot summarizes LISI values ($n = 16,274$ cells, ~50% of the cells in the dataset as suggested in ref. 24), the box denotes the interquartile range (IQR, the range between the 25th and 75th percentile) with the median value, whiskers indicate the maximum and minimum value within 1.5 times the IQR, outliers are denoted by black circles. **h** Qualitative assessment of batch-mixing by visualization of scDREAMER's latent space embeddings, cells are colored based on three categories—positive, negative and true positive. **i** Quantitative assessment of batch-mixing of scDREAMER against scVI and Harmony based on the percentage of positive vs true positive cells. Source data are provided as a Source Data file.

successful in clustering together the same cell types across tissues while identifying cell subtypes, such as CD8$^+$ and CD4$^+$ T cells, CD20$^+$ and CD10$^+$ B cells, CD14$^+$ and CD16$^+$ monocytes, as distinct closely located clusters. In comparison, CD20$^+$ and CD10$^+$ B cells were distantly placed in BBKNN and Scanorama embeddings (Supplementary Fig. 7e, f) and CD20$^+$ B cells were fragmented into two clusters in scDML embeddings (Supplementary Fig. 7j). While scDREAMER was able to place monocyte-derived dendritic cells and plasmacytoid dendritic cells in separate clusters in close vicinity (Fig. 4a, b), other methods such as Harmony,

Seurat, scDML and INSCT (Supplementary Fig. 7c, d, g) placed these cell subtypes far apart in the embedding. Furthermore, scVI separated monocyte-derived dendritic cells into multiple clusters, one of which was mixed with other distinct cell types (e.g., megakaryocyte progenitors, plasma cells, CD4$^+$ T cells, etc.) indicating poor clustering (Supplementary Fig. 7b). Finally, scDREAMER clearly separated tissue-specific cell types and exhibited a perfect continuum of bone marrow cell types after integration, preserving the trajectory from HSPCs to erythrocytes (Fig. 4a, b, Supplementary Fig. 8a, b). In contrast, methods

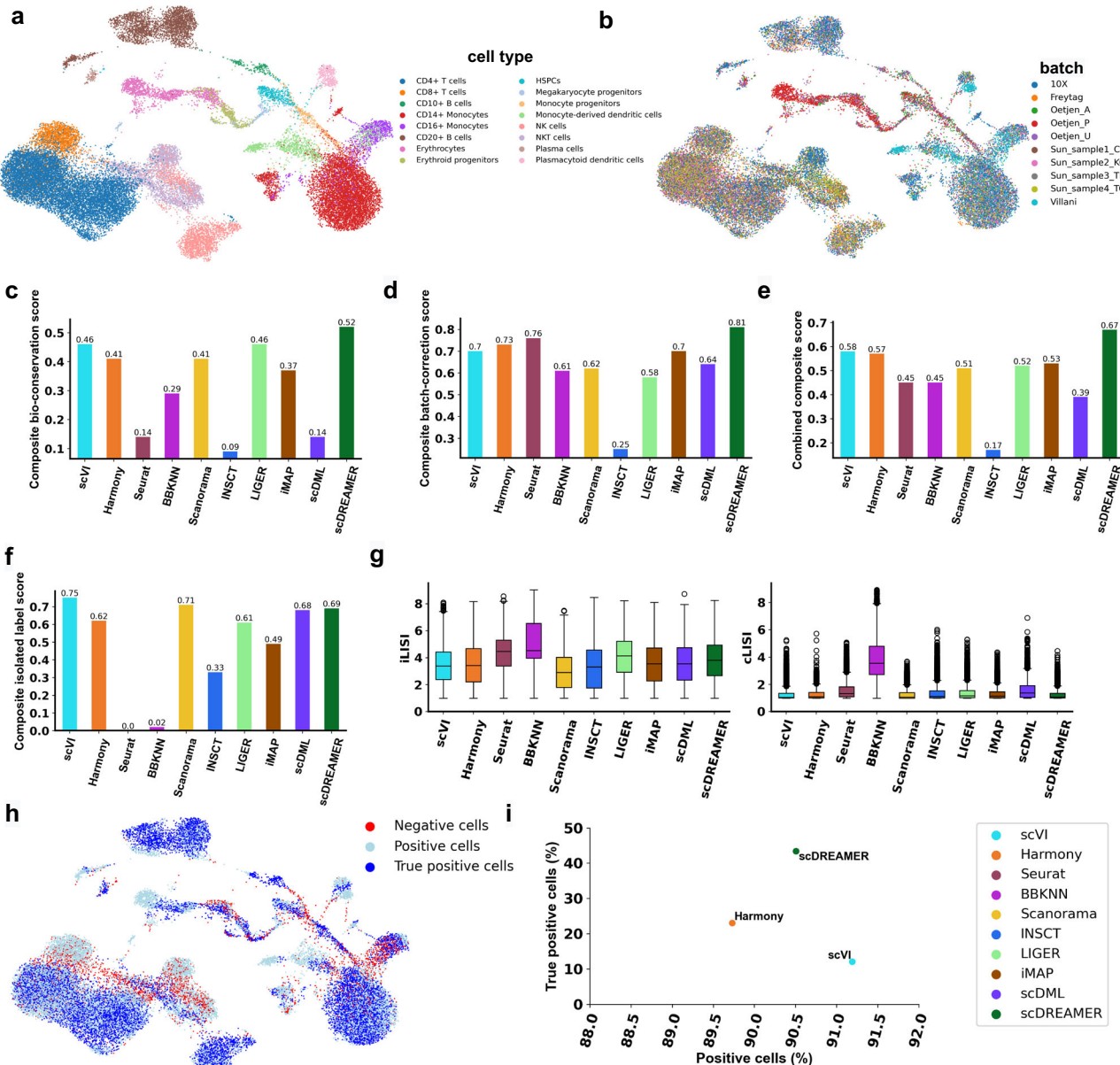

**Fig. 4 | Integration of human immune data. a** Visualization of scDREAMER's latent space embeddings after integration of human immune dataset. Different colors denote different cell types present in the human immune dataset. **b** Visualization of scDREAMER's latent space embeddings, cells are colored based on the batch information. Comparison of **c** composite bio-conservation score, **d** composite batch-correction score and **e** combined composite score metrics between scVI, Harmony, Seurat, BBKNN, Scanorama, INSCT, LIGER, iMAP, scDML and scDREA-MER for the integration of human immune data. **f** Comparison of composite iso-lated label scores to assess how well rare cell types are identified. **g** Comparison of iLISI and cLISI values. Each box-and-whisker plot summarizes LISI values ($n = 16,754$

cells, ~50% of the cells in the dataset as suggested in ref. 24), the box denotes the interquartile range (IQR, the range between the 25th and 75th percentile) with the median value, whiskers indicate the maximum and minimum value within 1.5 times the IQR, outliers are denoted by black circles. **h** Qualitative assessment of batch-mixing by visualization of scDREAMER's latent space embeddings, cells are colored based on three categories—positive, negative and true positive. **i** Quantitative assessment of batch-mixing of scDREAMER against scVI and Harmony based on the percentage of positive vs true positive cells. Source data are provided as a Source Data file.

such as Seurat, INSCT, LIGER and scDML (Supplementary Fig. 7d, g, h, j) failed to conserve the trajectory of erythrocyte development, whereas, iMAP divided the erythroid trajectory into three batch-specific trajec-tories (Supplementary Fig. 7i).

Our quantitative comparison showed that scDREAMER con-sistently outperformed all other methods by achieving the highest combined composite score (15.5% improvement over the second-best method) which was driven by scDREAMER's superior performance in both biological conservation and batch correction (Fig. 4c–e). scDREAMER was the best-performing method in terms of kBET, and

two bio-conservation metrics (Supplementary Fig. 9a, b). In capturing the rare cell identities, scDREAMER also performed at par with scVI and Scanorama, the top-performing methods in terms of composite iso-lated label scores (Fig. 4f, Supplementary Fig. 9c). However, for batch correction, scVI and Scanorama performed poorly. In terms of LISI metrics, scDREAMER along with scVI and Harmony consistently per-formed better than other methods (Fig. 4g). We further compared scDREAMER's performance against that of Harmony and scVI (second and third-best methods based on combined composite score respec-tively) by measuring the proportion of positive and true positive cells.

While both scDREAMER and scVI had a comparable proportion of positive cells better than Harmony, scDREAMER outperformed both methods by a large margin based on the proportion of true positive cells (Fig. 4h, i).

Finally, to evaluate scDREAMER's ability in uncovering unique biological insights, we focused on a more in-depth analysis of the dendritic cell populations after integration. Based on scRNA-seq and functional studies, Villani et al.[29] characterized six different subtypes of dendritic cells—DC1 (CLEC9A⁺), DC2 (non-inflammatory CD1C⁺), DC3 (inflammatory CD1C⁺), DC4 (CD141⁻CD1C⁻), DC5 (AXL⁺), and DC6 (pDCs). Through subclustering and marker gene expression analysis, we wanted to find out if an integration method was able to separate these DC subtypes after the integration of multiple batches. scVI performed poorly in identifying the DCs as a separate cluster, the optimal clustering of scVI embeddings (considering all cells) revealed that DC1 cells were clustered with CD20⁺ B cells and DC2 and DC3 cells were clustered with CD14⁺ monocytes (Supplementary Fig. 10a). Given its inability to identify the dendritic cells as a separate cluster, we did not perform any subclustering. In comparison, scDREAMER and Harmony were able to identify the dendritic cells as separate clusters. Subclustering of scDREAMER-inferred dendritic cell embeddings resulted in seven clusters each of which corresponded to distinct DC subtypes (Supplementary Fig. 10b). Clusters 0 and 1 contained DC2 and DC3 cells, clusters 2 and 3 contained pDCs whereas clusters 4, 5, and 6 contained DC1, DC4, and DC5 respectively. Thus, scDREAMER was able to identify 4 (out of 6) DC subtypes as separate clusters. In comparison, optimal subclustering of Harmony embeddings of DCs revealed 4 clusters (Supplementary Fig. 10c), out of which only two clusters corresponded to DC4 and pDCs. Harmony failed to properly distinguish other DC subtypes—DC2 and DC3 cells were mixed with DC4 cells, whereas, DC1 and DC5 cells were present across multiple clusters.

## scDREAMER-Sup utilizes available cell type labels for improved conservation of biological knowledge

Next, we evaluated scDREAMER-Sup's ability to leverage the available cell type labels for improved integration on lung atlas and human immune datasets as these integration tasks were the most challenging. We compared scDREAMER-Sup's performance against that of scANVI[30] and scGEN[31], state-of-the-art supervised methods (also top performers in recent benchmarking[24]) that utilize cell type labels. scDREAMER-Sup was able to capture all the cell types in distinct clusters for both integration tasks improving upon the performance of scDREAMER. Particularly, for lung atlas integration, scDREAMER-Sup was able to separate two neutrophil subtypes which were merged by all unsupervised methods (Fig. 5a, b). In comparison, integration by scANVI led to restricted separation of Macrophage and dendritic cells, and merging of neutrophil subtypes while scGEN mixed some B cells and T/NK cells as well as some ciliated and secretory cells (Supplementary Fig. 11a–d). In addition, B cells and secretory cells were fragmented in multiple closely located clusters by scANVI. When integrating human immune data, scGEN and scANVI could not properly separate CD14⁺ and CD16⁺ monocytes as well as NK cells and NKT cells (Supplementary Fig. 11e–h). scDREAMER-Sup was able to identify these cell subtypes as disjoint clusters (Fig. 5i, j).

We further quantitatively evaluated the performance of scDREAMER-Sup for both supervised (cell type labels available for all cells) as well as semi-supervised settings (cell type labels available for a fraction of cells). To evaluate the integration methods in semi-supervised settings, we uniformly randomly removed cell type labels for a percentage of cells from all the cell types while we varied the percentage of missing labels (10, 20, and 50%). For both datasets across all experimental settings, scDREAMER-Sup outperformed other methods based on combined composite (25–40% improvement over next best method), composite bio-conservation (14–33% improvement over next best method), and composite batch-correction scores

(21–49% improvement over next best method) (Fig. 5c–e, k–m). It is important to note that for both datasets, scDREAMER-Sup also outperformed all the unsupervised methods (scDREAMER was the best unsupervised method for both datasets) in terms of both bio-conservation (84.61–113.33% improvement over best unsupervised method) and batch-correction (16.05–19.72% improvement over best unsupervised method). With an increase in the percentage of missing labels, we observed a sharp drop in scGEN's composite bio-conservation score which is indicative of the fact that scGEN requires labels for all cells and cannot account for missing cell type labels. In comparison, the performance of scDREAMER-Sup and scANVI were more robust to missing cell type labels with scDREAMER-Sup outperforming scANVI by a large margin in terms of all composite scores.

scDREAMER-Sup's superior performance was driven by top scores in multiple bio-conservation (NMI, ARI, ASW) (Supplementary Figs. 12a and 13a) and batch-correction metrics (kBET, graph connectivity, and ASW label/batch) (Supplementary Figs. 12b and 13b) across the integration tasks and experimental settings. scDREAMER-Sup was further able to capture rare cell identities very well as indicated by its top performance in terms of isolated f1 score (Supplementary Figs. 12c and 13c). scDREAMER-Sup further outperformed the other methods in terms of LISI metrics (Fig. 5f, n) across the experimental settings. Furthermore, we compared the methods by measuring the proportion of positive and true positive cells for each experimental setting. For both supervised and semi-supervised settings, scDREAMER-Sup outperformed the other two methods by achieving a higher proportion of positive and true positive cells (Fig. 5g, o). Since both scDREAMER-Sup and scANVI can predict the labels of the cells for which cell type annotations are missing, we also compared the accuracy of the cell label prediction for these two methods. For all the semi-supervised settings with 10, 20 and 50% cells missing labels for both the datasets, scDREAMER-Sup achieved high accuracy (Macro F1-score 0.86–0.9) in predicting the cell type labels and outperformed scANVI (10–14% and 23–33% improvement over scANVI for lung atlas and human immune dataset, respectively) (Fig. 5h, p).

## scDREAMER outperforms other methods in integrating a large number of batches with complex batch effects

We next evaluated scDREAMER's (both unsupervised and semi-supervised) integration and batch correction on large number of batches (also large number of cells) using a human heart atlas dataset consisting of 486,134 cells spanning six heart regions namely apex, left atrium, left ventricle, right atrium, right ventricle, and septum (https://www.heartcellatlas.org/,[32]). All tissues were from transplant donors without a history of cardiac disease or arrhythmia. The cells were obtained in 147 batches corresponding to different donors, sequential protocols, race, gender, age range, and death-type conditions. In the original study[32], a total of 452,136 cells were assigned to 12 major cardiac cell types using state-of-the-art machine learning methods, and the remaining 33,998 cells were not assigned any cell type label (Supplementary Fig. 14a). In our analyses, we used all the cells for integration but the cells with available cell type labels were used for the evaluation. Only the available cell type labels were used for training supervised methods making it a semi-supervised integration task for them. Apart from a large number of batches, the integration task posed several challenges including variability across human donors with different genders (~55% and ~45% from male and female donors, respectively), race (93% and 6% cells from donors of Caucasian and Asian descent, respectively), and age group (spanning the age range of 40–75 years), sequencing protocol-specific batch effects, and cell types spanning different death-type conditions (265,554 cells (~40 batches) were sampled from the heart in case of circulatory death, and the rest 220,580 cells (~100 batches) were sampled from the heart in case of brain death). Furthermore, the dataset contained skewed cell type distribution, i.e., ~50 batches did not have any cell corresponding

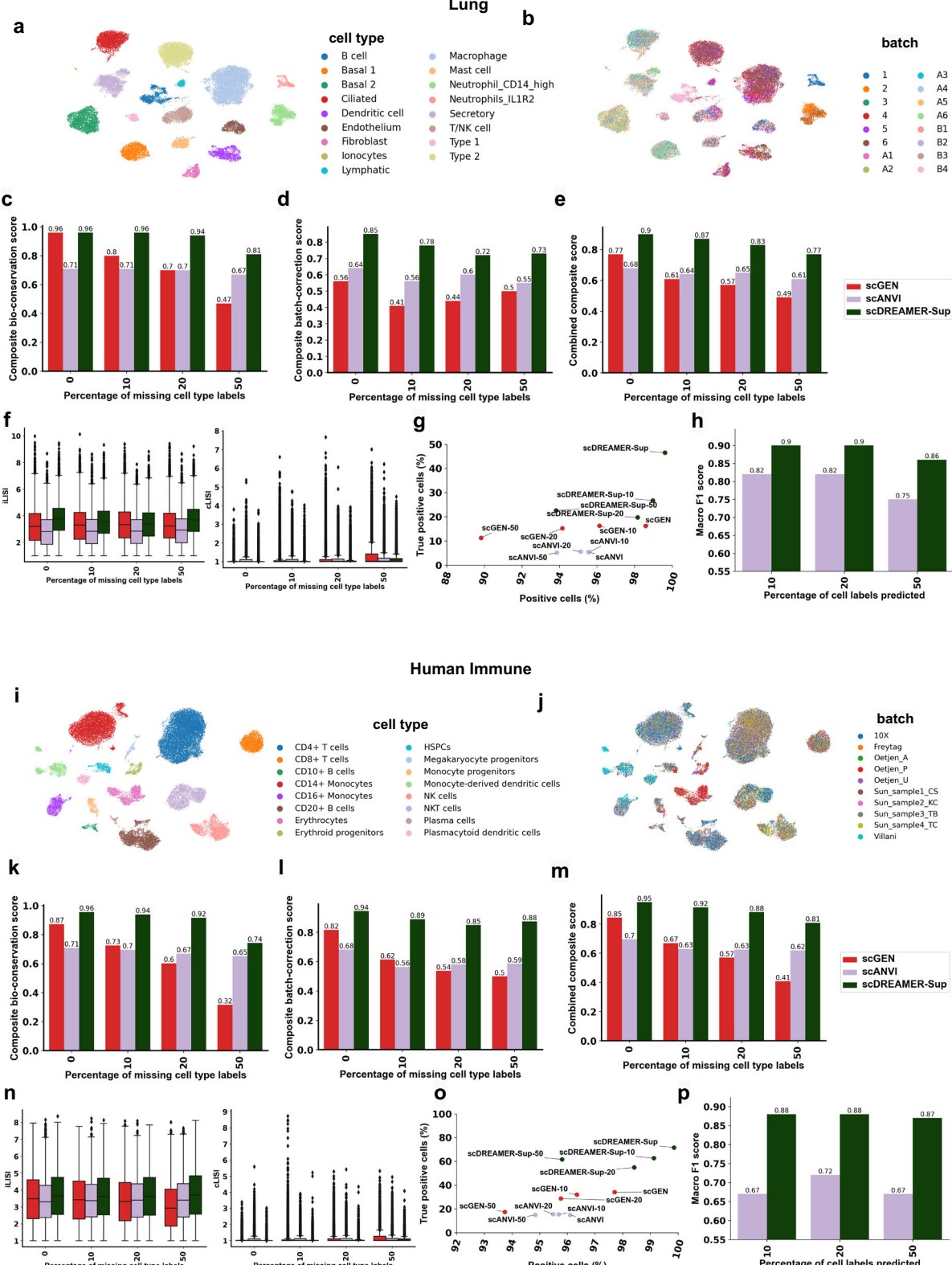

to Ventricular Cardiomyocyte and Fibroblast resulting in a multi-faceted challenging integration problem.

Among the unsupervised methods, Seurat, LIGER, scDML and iMAP failed to integrate this dataset given their huge memory requirement (tested on 512 GB memory) or inability to handle such a large number of batches. Most of the other methods that could integrate this dataset led to the mixing of atrial and ventricular cardio-myocytes given their similarity. BBKNN and Scanorama fragmented

many cell types into multiple clusters (Supplementary Fig. 14d, e). scVI, Harmony, and scDREAMER were able to capture most cell types into distinct clusters (Fig. 6a, b, Supplementary Fig. 14b, c). However, scVI and Harmony also led to the fragmentation of smooth muscle cells and poor batch correction for Lymphoid cells. In comparison, supervised methods were able to better capture the different cell types in distinct clusters by leveraging the available cell type labels (Fig. 6c, d, Sup-plementary Fig. 14g, h). However, scANVI still failed to separate atrial

**Fig. 5 | scDREAMER-Sup utilizes cell type labels to improve bio-conservation.**
**a** Visualization of scDREAMER-Sup's latent space embeddings after integration of lung atlas dataset. Different colors denote different lung cell types. **b** Visualization of scDREAMER-Sup's latent space embeddings, cells are colored based on the batch information. Comparison of **c** composite bio-conservation score, **d** composite batch-correction score, **e** combined composite score metrics, **f** iLISI and cLISI values between scGEN, scANVI, and scDREAMER-Sup for different percentages of missing cell type labels for lung atlas dataset. **g** Quantitative assessment of batch-mixing of scDREAMER-Sup against scANVI and scGEN based on the percentage of positive vs true positive cells for lung atlas dataset. **h** Comparison of cell label prediction accuracy between scDREAMER-Sup and scANVI for different percentages of missing cell type labels for the lung atlas dataset. **i** Visualization of scDREAMER-Sup's latent space embeddings after integration of human immune dataset. Different colors denote different lung cell types. **j** Visualization of scDREAMER-Sup's latent space embeddings, cells are colored based on the batch

information. Comparison of **k** composite bio-conservation score, **l** composite batch-correction score, **m** combined composite score metrics, **n** iLISI and cLISI values between scGEN, scANVI, and scDREAMER-Sup for different percentages of missing cell type labels for the human immune dataset. **o** Quantitative assessment of batch-mixing of scDREAMER-Sup against scANVI and scGEN based on the percentage of positive vs true positive cells for the human immune dataset. **p** Comparison of cell label prediction accuracy between scDREAMER-Sup and scANVI for different percentages of missing cell type labels for the human immune dataset. For (**f**) and (**n**), each box-and-whisker plot summarizes LISI values for 50% of the cells in the datasets as suggested in ref. 24 ((**f**) $n = 16,260$ cells, (**n**) $n = 16,770$ cells), the box denotes the interquartile range (IQR, the range between the 25th and 75th percentile) with the median value, whiskers indicate the maximum and minimum value within 1.5 times the IQR, outliers are denoted by black circles. Source data are provided as a Source Data file.

and ventricular cardiomyocytes (Supplementary Fig. 14h). In addition, scANVI also fragmented smooth muscle cells into multiple clusters. On the other hand, scGEN poorly handled the unassigned cells given its inability to handle missing cell type labels (Supplementary Fig. 14g). scDREAMER-Sup was able to resolve the complex batch effects while integrating heart cells as indicated by disjoint well-mixed cell type clusters (Fig. 6c, d). Both scDREAMER and scDREAMER-Sup were able to mix the cells well across tissues, protocols, donors, genders, age groups, and death-type conditions (Supplementary Fig. 15).

Quantitative analyses showed that scDREAMER performed the best among the unsupervised methods based on the combined composite score (25% improvement over the second-best method) which was driven by scDREAMER's superior performance in batch correction (~50% improvement over the second-best method) as well as good performance in bio-conservation (Fig. 6e–g, Supplementary Fig. 16). We could not include BBKNN in the comparison of the composite scores due to the unavailability of the majority of the metrics. However, scDREAMER outperformed BBKNN on both bio-conservation metrics (Supplementary Fig. 17a). Moreover, we could run scDML on this dataset after the removal of 7 batches as suggested in ref. 33 and in integrating the remaining 140 batches, scDREAMER outperformed scDML in terms of most of the metrics (Supplementary Fig. 17b). Among the supervised methods, scDREAMER-Sup achieved the best combined composite score (11.5% improvement over the second-best supervised method and 26% improvement over the best unsupervised method (Fig. 6e–g)). We further evaluated scDREAMER-Sup's performance in semi-supervised settings with 20 and 50% missing labels. In both settings, scDREAMER-Sup significantly outperformed scANVI and scGEN in both bio-conservation and batch correction (32–57% improvements, Fig. 6h–j, Supplementary Fig. 18a, b). scGEN's performance dropped sharply with an increase in the percentage of missing labels. scDREAMER-Sup and scANVI were robust to missing cell type labels with scDREAMER-Sup outperforming scANVI by a large margin in terms of all composite scores. Isolated label scores could not be computed for this dataset due to the sheer imbalance of batch sizes. We also compared the accuracy of the cell label prediction for scDREAMER-Sup and scANVI. For both the semi-supervised settings with 20 and 50% cells missing labels, scDREAMER-Sup achieved very high accuracy (macro F1-score 0.97) in predicting the cell type labels and outperformed scANVI (11.49% improvement over scANVI) (Fig. 6k). We further compared scDREAMER's performance against that of Harmony and scVI (second and third-best unsupervised methods based on combined composite scores respectively) by measuring the proportion of positive and true positive cells. While all three methods had a comparable proportion of positive cells, scDREAMER outperformed both methods by a large margin based on the proportion of true positive cells (Fig. 6l, m). Finally, we compared the supervised methods by measuring the proportion of positive and true positive cells for each experimental setting. For all experimental settings, scDREAMER-Sup

outperformed the other two methods by achieving a higher proportion of positive and true positive cells (Fig. 6n, o).

We further analyzed a macaque retina bipolar cells dataset consisting of 30,302 cells that originated from the fovea or periphery of the retina and consisted of 30 batches across macaques and regions. The original study showed that several integration methods failed to remove batch effects from this dataset[34]. Both scDREAMER and scDREAMER-Sup successfully integrated all 30 batches while separating the distinct cell types (Supplementary Fig. 19). In contrast, many other methods mixed some distinct cell types or fragmented same cell type into multiple clusters (Supplementary Fig. 20). Based on quantitative analyses also, scDREAMER-Sup was the best performer among all unsupervised and supervised methods in terms of multiple bio-conservation and batch correction metrics (Supplementary Fig. 21). Among the unsupervised methods, scDREAMER was the best performer based on NMI, ARI and kBET and performed comparably on all other metrics except ASW label. scDREAMER-Sup was also superior in identifying the rare cell types (Supplementary Fig. 21).

## scDREAMER robustly integrates millions of cells across different species

We finally evaluated scDREAMER's ability to perform atlas-level integration across species using a dataset consisting of ~1 million cells from human and mouse profiled by the Human Cell Landscape (HCL)[35] and Mouse Cell Atlas (MCA)[36] projects respectively. The two batches in the dataset correspond to HCL and MCA respectively. The dataset harbored 97 different cell types which had minimal overlap across the atlases (Supplementary Fig. 22a). The HCL batch comprised 599,926 cells from 63 different cell types whereas the mouse cell atlas comprised 333,778 cells from 52 different cell types. A total of 18 cell types (among the 97 cell types) were common between these two atlases. We have demonstrated scDREAMER's scalability over a million cells and versatility in integration with this atlas-level integration task.

For this atlas-level integration task, the performance of scDREAMER and scDREAMER-Sup was compared with that of eight other unsupervised methods (scVI, Scanorama, Harmony, INSCT, BBKNN, LIGER, iMAP and scDML) and two supervised methods (scGEN and scANVI). Due to the size of the dataset, Seurat ran into memory issues. Qualitatively, we can observe that clusters by scDREAMER were better separated compared to other methods (Fig. 7a, b, Supplementary Fig. 22b–i). scDREAMER was able to clearly distinguish some of the major cell types such as neutrophil cells, erythroid cells, fetal stromal cells and oligodendrocyte cells (Fig. 7a). While scVI could cluster some of these cell types such as neutrophil, erythroid and fetal stromal cells well, harmony could only cluster erythroid cells. In Harmony embeddings, similar cell types were disjointly present in multiple clusters. On the other hand, BBKNN and LIGER led to the complete mixing of multiple cell types. While Scanorama was able to better classify than BBKNN and Harmony, it was not able to properly mix the batches

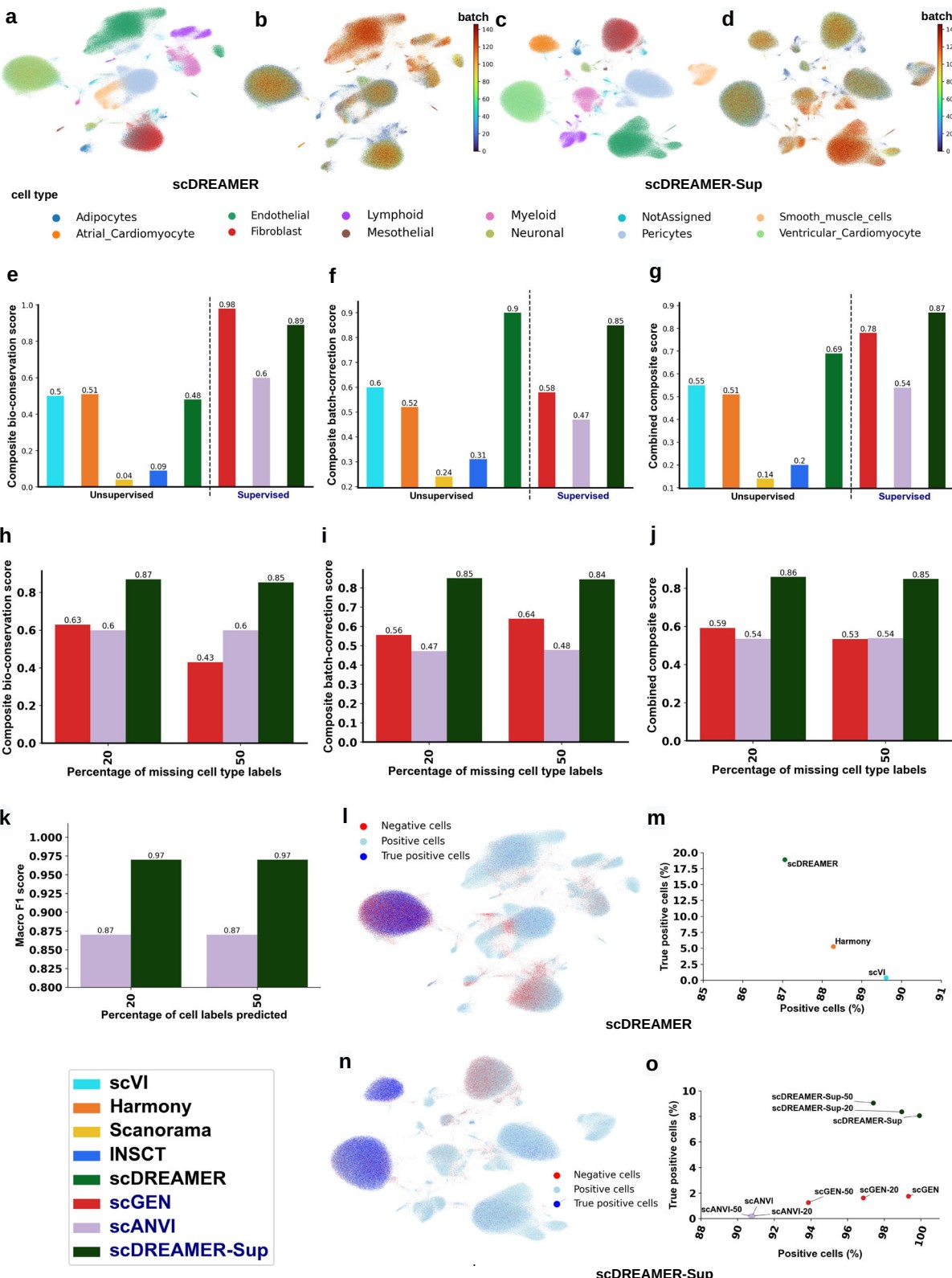

(Supplementary Fig. 22e). iMAP was also not able to mix the batches properly (Supplementary Fig. 22h). scDML mixed multiple cell types and fragmented some of the cell types into multiple clusters (Supplementary Fig. 22i). Among the supervised methods, scDREAMER-Sup and scANVI led to better-separated clusters, whereas scGEN was not able to cluster the cell types properly (Fig. 7c, d, Supplementary Fig. 22j, k).

In quantitative comparison, scDREAMER achieved the best combined composite score among all the unsupervised methods (26.9% better than the second-best unsupervised method) and was even better than the supervised methods scANVI and scGEN, whereas overall, scDREAMER-Sup had the best combined composite score (8.75% better than that of scDREAMER) (Fig. 7e–g). While scDREAMER was the best batch correction method, scDREAMER-Sup achieved the

**Fig. 6 | scDREAMER integrates heart atlas cells from a large number (147) of batches. a** Visualization of scDREAMER's latent space embeddings after the integration of 147 batches. Different colors denote different cell types in this large dataset consisting of ~0.5 million cells. 'NotAssigned' represents the cells without any cell type assignment. **b** Visualization of scDREAMER's latent space embeddings, cells are colored based on the batch information. **c** Visualization of scDREAMER-Sup's latent space embeddings, cells are colored based on cell types. **d** Visualization of scDREAMER-Sup's latent space embeddings, cells are colored based on the batch information. Comparison of **e** composite bio-conservation score, **f** composite batch-correction score, and **g** combined composite score metrics between unsupervised (scVI, Harmony, Seurat, Scanorama, INSCT, scDREAMER) and supervised (scGEN, scANVI, and scDREAMER-Sup) methods. Comparison of **h** composite bio-conservation score, **i** composite batch-correction score, and **j** combined composite score metrics between scGEN, scANVI and scDREAMER-Sup for different

percentages of missing cell type labels for the heart atlas dataset. **k** Comparison of cell label prediction accuracy between scDREAMER-Sup and scANVI for different percentages of missing cell type labels for the heart atlas dataset. **l** Qualitative assessment of batch-mixing by visualization of scDREAMER's latent space embeddings, cells are colored based on three categories–positive, negative and true positive. **m** Quantitative assessment of batch-mixing of scDREAMER against that of scVI and Harmony based on the percentage of positive vs. true positive cells. **n** Qualitative assessment of batch-mixing by visualization of scDREAMER-Sup's latent space embeddings, cells are colored based on three categories–positive, negative and true positive. **o** Quantitative assessment of batch-mixing of scDREAMER-Sup against that of scANVI and scGEN based on the percentage of positive vs. true positive cells for the heart atlas dataset. Source data are provided as a Source Data file.

best bio-conservation (best in terms of all bio-conservation metrics) (Supplementary Fig. 23). We further compared scDREAMER's performance against that of scVI (second-best unsupervised method), Scanorama and LIGER (third-best unsupervised method) and scDREAMER-Sup's performance against that of scGEN and scANVI by measuring the proportion of positive and true positive cells (Fig. 7h–k). scDREAMER outperformed Scanorama and LIGER based on the proportion of positive cells and all the unsupervised methods based on the proportion of true positive cells (Fig. 7h). On the other hand, based on both the proportion of positive and true positive cells, scDREAMER-Sup outperformed scGEN and scANVI by a large margin. (Fig. 7i).

To assess scDREAMER's scalability with the number of cells, we measured the runtime of scDREAMER on different datasets subsampled from the atlas-integration dataset and compared the runtime against that of scVI and INSCT–two other neural network-based methods. For most of the subsampled datasets, scDREAMER outperformed the other two methods based on runtime and exhibited higher scalability with the number of cells (Fig. 7l). For the complete dataset consisting of ~1 million cells, training scDREAMER took only ~130 s per epoch as compared to ~154 s per epoch for scVI and ~172 s per epoch for INSCT (on a server with one Nvidia Quadro RTX 5000 GPU). We further compared the total runtime of different methods across different datasets (Supplementary Fig. 24). scDREAMER was one of the fastest among all the neural network-based methods across all datasets and was faster compared to some other methods (e.g., Seurat, LIGER). scDREAMER-Sup was also faster than scANVI for multiple datasets and faster than other neural network-based unsupervised methods.

## Ablation study

We performed ablation studies to examine the contribution of the two adversarial components of scDREAMER–adversarial discriminator and batch classifier. Supplementary Table 2 compares the performance of scDREAMER in comparison to scDREAMER without the discriminator (scDREAMER-woDis) and scDREAMER without the batch classifier (scDREAMER-woBC) for different integration tasks. As can be seen, scDREAMER-woBC performs better than scDREAMER-woDis in terms of composite bio-conservation score and composite isolated label score indicating the contribution of the discriminator toward improved bio-conservation and identification of rare cell types. In contrast, the version that uses batch classifier (scDREAMER-woDis) performs better in terms of batch correction as compared to scDREAMER-woBC indicating the contribution of the batch classifier in removing batch effects. Thus, both the components play a significant role and by combining them, scDREAMER achieves the best performance in terms of combined composite score as well as composite isolated label score for different datasets.

## Discussion

Here, we introduced scDREAMER, a deep generative model for the efficient and robust integration of scRNA-seq datasets across multiple

batches. The unsupervised model of scDREAMER employs an adversarial variational autoencoder for inferring the latent cellular embeddings from the high-dimensional gene expression matrices from different batches. This adversarial autoencoder also outputs the corrected expression profiles. The other component of scDREAMER, a batch classifier helps remove batch effects from the latent cellular embeddings for better mixing of cell types shared across multiple batches. Thus, scDREAMER employs a combination of two levels of adversarial training for training the adversarial VAE and the batch classifier respectively and differs from existing adversarially trained deep generative models for dimensionality reduction[37] and batch integration of scRNA-seq data[14,22] (see Supplementary Note 1 for details). Our ablation study further demonstrated that the two adversarially trained components play important roles in improving bio-conservation and removal of batch effects respectively. We further extended scDREAMER to scDREAMER-Sup, which employs an additional variational autoencoder and a cell type classifier to utilize available cell type labels for improved bio-conservation.

Our comprehensive benchmarking of scDREAMER and scDREAMER-Sup on multiple complex data integration tasks demonstrates scDREAMER's superiority over the state-of-the-art unsupervised and supervised integration methods respectively in terms of both conservation of biological variations and removal of batch effects. In contrast, the other methods were only able to perform well in one aspect of data integration: batch-mixing or conservation of biological variation, that too varied across datasets. scDREAMER was also a consistent performer in capturing rare cell identities. Moreover, a comparison of the fraction of true positive and positive cells further demonstrated scDREAMER's superiority over other methods in batch-mixing. scDREAMER-Sup was found to improve upon scDREAMER and it consistently outperformed all the supervised (that utilize the cell type annotations) as well as unsupervised integration methods across various benchmarking datasets and experimental settings.

scDREAMER also demonstrated high accuracy for the integration of a large number of batches and atlases from different species despite the small number of shared cell types. Both scDREAMER and scDREAMER-Sup significantly outperformed the other methods for the integration of heart atlas across 147 batches, which many other methods failed to integrate. scDREAMER-Sup also performed superior to all other methods in predicting the cell type labels for the cells missing annotations. Moreover, the unsupervised deep learning approach of scDREAMER does not require any cell type information and can be applied when prior knowledge regarding homologous cell types is not available or the cell type annotations are missing. In case cell type labels are available (completely or partially), the supervised version of scDREAMER, scDREAMER-Sup can achieve improved performance in bio-conservation while maintaining batch-effect removal performance. As more cell atlases are generated from different species, we believe that scDREAMER will be suitable for robust integration of cross-species datasets for the discovery of shared and private cell types.

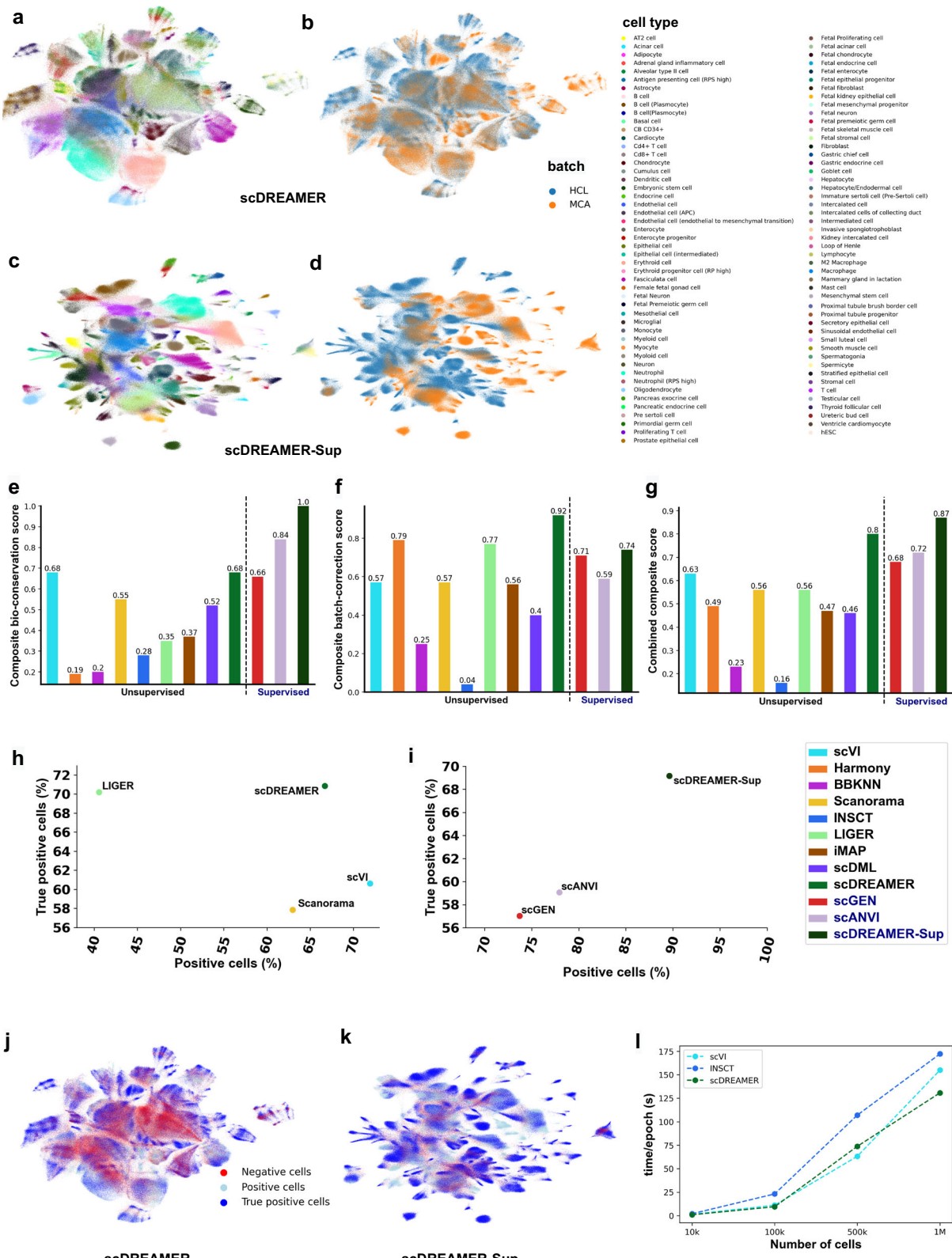

The application of scDREAMER on the heart atlas and cross-species dataset also highlights its scalability to a large number of batches and millions of cells. In fact, our runtime experiments using different down-sampled versions of the cross-species dataset showed that scDREAMER outperformed some other deep learning-based methods based on runtime. Thus scDREAMER provides a suitable deep learning-based integration model for cross-species atlas integration as the other deep learners scVI, INSCT, iMAP and scDML achieved much less accuracy for the cross-species integration task. This is particularly important as the deep learning-based methods enable the inference of latent cellular embeddings as well as corrected expression profiles which are required for several downstream applications such as trajectory inference[10] or differential expression analysis[20].

**Fig. 7 | scDREAMER enables robust integration of millions of cells across species. a** Visualization of scDREAMER's latent space embeddings after the integration of human (HCL) and mouse cells (MCA). Different colors denote different cell types in this large dataset consisting of ~1 million cells. **b** Visualization of scDREAMER's latent space embeddings, cells are colored based on the batch information. **c** Visualization of scDREAMER-Sup's latent space embeddings after the integration of human (HCL) and mouse cells (MCA). Different colors denote different cell types. **d** Visualization of scDREAMER-Sup's latent space embeddings, cells are colored based on the batch information. Comparison of **e** composite bio-conservation score, **f** composite batch-correction score and **g** combined composite score metrics between scVI, Harmony, Seurat, BBKNN, Scanorama, INSCT, LIGER, iMAP, scDML, scDREAMER, scGEN, scANVI and scDREAMER-Sup for the integration of HCL and MCA cells. **h** Quantitative assessment of batch-mixing of scDREAMER against that of scVI, Scanorama and LIGER based on the percentage of positive vs true positive cells. **i** Quantitative assessment of batch-mixing of scDREAMER-Sup against that of scGEN and scANVI based on the percentage of positive vs true positive cells. **j** Qualitative assessment of batch-mixing by visualization of scDREAMER's latent space embeddings, cells are colored based on three categories −positive, negative and true positive. **k** Qualitative assessment of batch-mixing by visualization of scDREAMER-Sup's latent space embeddings, cells are colored based on three categories−positive, negative and true positive. **l** Comparison of scDREAMER runtime against that of scVI and INSCT across four different scRNA datasets consisting of 10k, 100k, 500k, and 1M cells subsampled from the cross-species integration dataset. Source data are provided as a Source Data file.

scDREAMER being a deep learning-based method may require fine-tuning of some parameters for extracting the best performance. However, in our analyses, the same parameter values performed well across multiple datasets (Supplementary Table 3). Our current model assumes prior knowledge of the number of batches. An important future direction would be to explore the unsupervised treatment of batch information and whether the hierarchical structure between different batch information can be utilized when it exists (e.g., when cells from a single donor but multiple organs are present in the atlas). While we restricted our analysis to the integration of scRNA-seq datasets, our deep generative model encompasses a general framework that can accommodate other omics datasets and we plan to extend the framework of scDREAMER to multiomic datasets. Finally, given the rapid generation of atlas-level single-cell datasets[38–40] across multiple organs and species, we anticipate supervised and unsupervised models of scDREAMER to become invaluable methods for performing scalable and accurate integration of single-cell atlases for the exploration of different biological systems.

## Methods
### The scDREAMER model
The unsupervised model of scDREAMER consists of an unsupervised deep learning-based framework specifically designed to address the complex and multi-level batch effects and perform atlas-level integration ensuring effective integration as well as batch mixing. Any integration method is faced with the critical challenge of balancing a tradeoff between capturing the distinct identity of the batch-specific cell types and adequate mixing of the cell types shared across multiple batches. This challenge is even more profound for the integration of atlas-level datasets. To overcome these challenges, we formalize our scDREAMER integration model into two major components, the adversarial variational autoencoder and the batch classifier that is adversarially trained with the encoder.

scDREAMER employs an adversarial variational autoencoder for learning a lower-dimensional representation of cells from the high-dimensional scRNA-seq data. In addition, there is a neural network classifier (also called a batch classifier) for the removal of batch effects. The adversarial variational autoencoder of scDREAMER consists of three multi-layer neural networks: an encoder $E$ that maps the high-dimensional expression data ($x_i$) and batch information ($s_i$) of a cell $i$ to a lower-dimensional embedding $z_i$, a decoder $D$, which reconstructs the expression profile of the cell from $z_i$ and $s_i$, and a discriminator $\mathcal{D}$ that aims to distinguish the original expression profile $x_i$ and the expression profile reconstructed ($\bar{x}_i$) by the decoder. The use of a discriminator to adversarially train the autoencoder is inspired by ref. 37.

The adversarial variational autoencoder network of scDREAMER is trained using two loss functions: evidence lower bound (ELBO) is used for training the encoder and decoder networks, whereas Bhattacharyya loss is used for adversarial training of discriminator and autoencoder parameters. The ELBO loss accounts for the KL divergence between the posterior distribution $q(z_i|s_i, x_i)$ and the true distribution $p(z_i)$, and the expected likelihood of $x_i$ given $z_i$ over posterior probability distribution $q(z_i|s_i, x_i)$. The Bhattacharyya loss compares the probability distributions $q(z_i|x_i, s_i)$ and $p(z_i)$ where $q(z_i|x_i, s_i)$ is the posterior probability distribution and $p(z_i)$ is 0-mean Gaussian distribution, i.e., $\mathcal{N}(0, I)$.

scDREAMER further incorporates a batch classifier, $\mathcal{B}$ (a multi-layer neural network) that takes as input the lower-dimensional embedding $z_i$ learned by the encoder and tries to predict the batch information $s_i$ for each cell $i$. The batch classifier and the encoder are adversarially trained using a cross-entropy loss where the encoder tries to maximize it with an aim to generate the embeddings such that the classifier is not able to differentiate between batches and the batch classifier tries to minimize it by distinguishing the embeddings of the cells that are part of different batches and hence achieving better mixing of the batches.

### Adversarial variational autoencoder for representation learning of cells
$E$ denotes the Encoder network that takes as input $X$ (gene expression matrix) and $S$ (set of batch information) and generates the mean $\mu_Z$ and variance $\sigma_Z$ of the multivariate normal distribution (prior for $z$). In addition, it also learns the mean and variance of the cell-specific scaling factor $l_i$ as used in ref. 20.

The Encoder $E$ learns the functional form: $z_i \sim Normal(0, I)$, $f_E: \{x, s\} \rightarrow \{\mu_z, \sigma_z, \mu_l, \sigma_l\}$. The decoder network $D$ takes as input the latent space embeddings $z$ and batch information and reconstructs the gene expression matrix, $\bar{X}$. It also outputs the mean and dispersion of the reconstructed gene expression vector. The scRNA-seq data can be modeled as zero-inflated negative binomial (ZINB) or negative binomial (NB) distribution.

scDREAMER aims to learn the posterior distribution of the latent variable $z$, $p(z|X, S, l)$. We use a variational inference approach where we try to learn the network parameters of $E$ and $D$ as denoted by $\phi = \{f_E, f_D\}$ by maximizing the Evidence Lower Bound Loss (ELBO) function:

$$ELBO = -KL(q_\phi(z|x,s)||p(z)) - KL(q_\phi(l|x,s)||p(l)) + E_{q_\phi(z,l|x,s)} log(p(x|z,s,l))$$
(1)

where, the first two terms denote the KL divergence between the posterior distribution and true distribution for $z$ and $l$ respectively and the third term denotes the expected likelihood of $x$ given $z$ and $l$ over the posterior probability distribution, $q_\phi(z, l|x, s)$. In this work, we have modeled scRNA-seq data using ZINB distribution which is defined in terms of mean ($\mu$) and dispersion ($\theta$) parameters as:

$$\forall x \in \mathbb{N}; p(y; \mu, \theta) = \frac{\Gamma(x+\theta)}{\Gamma(x+1)\Gamma(\theta)} \left(\frac{\theta}{\theta+\mu}\right)^\theta \left(\frac{\mu}{\mu+\theta}\right)^x$$
(2)

**Training of discriminator.** Inspired by ref. 37, scDREAMER autoencoder also incorporates an adversarial discriminator $\mathcal{D}$ that tries to distinguish between the reconstructed gene expression profile $\bar{x}$ and the original gene expression profile $x$ of a cell. This ensures that the distribution of the reconstructed gene expression profiles faithfully

follows the distribution of the underlying original scRNA-seq profiles. $\mathcal{D}$ is adversarially trained along with the autoencoder, $\mathcal{A}_E = \{E, D\}$ using Bhattacharyya loss, $\mathcal{L}_\mathcal{B}$ given by:

$$\mathcal{L}_\mathcal{B} = -log\left(\sum_{i=1}^{N}\sqrt{P(x_i) \times P(\bar{x}_i)}\right) \tag{3}$$

$\mathcal{D}$ tries to maximize $\mathcal{L}_\mathcal{B}$ whereas the autoencoder $\mathcal{A}_E$ tries to minimize $\mathcal{L}_\mathcal{B}$ giving rise to the minimax objective function:

$$\min_{\mathcal{A}_E} \max_{\mathcal{D}} \mathcal{L}_\mathcal{B}(\mathbb{E}_{x \sim P_x}\mathcal{D}(x), \mathbb{E}_{z \sim p(z)}\mathcal{D}(\mathcal{A}_E(z))) \tag{4}$$

**Adversarial training using a batch classifier for batch-effect removal.** The batch-classifier network $\mathcal{B}$ learns its parameters by minimizing the cross-entropy loss thereby ensuring correct classification of embeddings into different batches while Encoder tries to fool the batch-classifier by maximizing the cross-entropy loss. $\mathcal{B}$ takes $z$ (latent space embeddings) as input and $s$ (batch information) as labels. The cross-entropy loss is given by:

$$-\sum_{j=1}^{|S|} s_j \times log(r_j) \tag{5}$$

where $s_j$ is the true label and $r_j$ is the softmax probability of the $jth$ batch.

The batch-classifier intends to minimize the cross-entropy loss, hence classifying the latent cellular embeddings to its correct batch whereas the encoder tries to maximize the cross-entropy loss ensuring that the classifier is not able to differentiate between batches so that better mixing of data from different batches can be achieved. This gives rise to another minimax objective:

$$\max_E \min_\mathcal{B} -\sum_{j=1}^{|S|} s_j \times log(r_j) \tag{6}$$

**Implementation details**

- scDREAMER has 4 neural network units correspnding to Encoder, Decoder, Discriminator and Batch-Classifier, all with multi-layer dense neural network architecture with linear relu activation at the end of each layer. For training our model, we have adopted ADAM optimizer[41]. We have used $\beta$ as a scaling factor (multiplied to the KL-divergence term) to balance between the reconstruction loss and KL-divergence term while optimizing the ELBO loss. The batch size of 128 is used while training the model.
- To facilitate and stabilize the training process and make scDREAMER robust to small perturbations, we added a penalty term in the main objective function following ref. [42]. Specifically, for each gene expression vector $x_i$, we down-sample $x_i$ by keeping 80% of its UMIs to produce $x_i$. The latent representations for $x_i$ and $\hat{x}_i$ are $z_i$ and $\hat{z}_i$ respectively. The penalty term is defined as $-\sum_{j=1}^{|z_i|}(z_{ij} - \hat{z}_{ij})^2$ as we want $z_i$ and $\hat{z}_i$ to be as close as possible. The down-sampling ratio used in our case is 80%.
- In our evaluations across different integration tasks and experimental settings, we have used the same values for hyperparameters and the same architecture for our neural networks. We further performed a robustness study for two learning rate parameters for Pancreas data integration (Supplementary Table 4), which showed that the metrics were comparable across different values of learning rate parameters.

**Extension of scDREAMER model for utilizing cell type labels**

The scDREAMER model was further extended to scDREAMER-Supervised (scDREAMER-Sup), which can leverage existing cell type

annotations to further improve biological conservation while integrating scRNA-seq datasets (Fig. 1). Inspired by the deep generative semi-supervised learner[43] and scANVI[30], scDREAMER-Sup employs a hierarchical structure for the inference of an informative latent embedding $z$ guided by the available cell type annotations $c$. Essentially, the prior on $z_i$ becomes more informative as it becomes conditioned on cell type label $c_i$ and another latent variable $y_i$ that accounts for within cell type variability. scDREAMER-Sup employs an additional variational autoencoder, $\mathcal{A}_Y = \{E_y, D_y\}$ for learning $y_i$ from $z_i$ and $c_i$, and another feed-forward neural network (also called cell type classifier), $\mathcal{C}$, for learning $c_i$ from $z_i$. When cell type annotations are available, those are fed to the autoencoder $\mathcal{A}_Y$ along with the latent embeddings $z_i$ from the adversarial variational autoencoder, $\mathcal{A}_E$. In case cell type annotations are partially available, scDREAMER-Sup learns the missing cell type labels by training the cell type classifier network using a cross-entropy loss on the set of cells for which cell type annotations are available.

**Training cell type classifier.** The cell type classifier network, $\mathcal{C}$ learns its parameters by minimizing the cross-entropy loss thereby ensuring correct classification of latent embeddings $z$ into different cell type annotations. $\mathcal{C}$ takes as input $z_i$ from $\mathcal{A}_E$ and $c_i$ (cell type annotation) as label and intends to minimize cross-entropy loss, hence classifying the latent cellular embeddings to its correct cell type. The cross-entropy loss is given by:

$$-\sum_{i=1}^{|C|} c_i \times log(r_i) \tag{7}$$

where $c_i$ is the true cell type label, $r_i$ is the softmax probability of the $i^{th}$ cell type annotation and $|C|$ is the number of cell types.

Under semi-supervised setting, $\mathcal{C}$ is trained using the labeled data, i.e., the cross-entropy loss is computed using the available cell type annotations. The trained network is then used to predict the cell type annotations for the unlabeled cells.

**Variational autoencoder for learning informed prior on cellular latent space.** The variational autoencoder $\mathcal{A}_Y$ consists of an encoder $E_y$ that takes as input $z_i$ and $c_i$ and generates the mean $\mu_y$ and variance $\sigma_y$ of the multivariate normal distribution (prior for $y$). The decoder network $D_y$ takes as input $y_i$ and $c_i$ and reconstructs latent cellular embedding $\bar{z}_i$. We assume the following distributions for $y_i$ and $z_i$

$$y_i \sim Normal(0, I), z_i \sim Normal(f_z'^\mu(y_i, c_i), f_z'^\sigma(y_i, c_i)) \tag{8}$$

where $f_z'^\mu$ and $f_z'^\sigma$ are two functions approximated by the variational autoencoder $\mathcal{A}_Y$.

We use $\phi = \{f_E, f_D\}$, and $\phi' = \{f_{E_y}, f_{D_y}\}$ to denote the network parameters of the two variational autoencoders ($\mathcal{A}_E$ and $\mathcal{A}_Y$ respectively) of scDREAMER-Sup which we learn using variational inference by maximizing an ELBO function given by:

$$\begin{aligned} ELBO = \ & E_{q_\phi(z,l|x,s)}\left[log\, p(x|z,l,s)\right] - KL(q_\phi(z|x,s)\,||\,E_{q_{\phi'}(y|z,c)}p(z|y,c)) \\ & -E_{q_\phi(z|x,s)}KL(q_{\phi'}(y|z,c)\,||\,p(y)) - KL(q_\phi(l|x,s)\,||\,p(l)) \end{aligned} \tag{9}$$

equation (9) denotes the ELBO function for only one cell (this is without loss of generality as the cells are independent and identically distributed) and assumes the following factorization of the variational distribution

$$q(z,y,l|x,s) = q(z|x,s)q(y|z,c)q(l|x,s) \tag{10}$$

The ELBO is derived following semi-supervised variational autoencoder literature[43]. Our variational distribution does not approximate the

posterior for cell type annotation $c$, instead for the cases, where $c$ is unavailable, we use the cell type classifier to generate the cell type annotations.

**Implementation details of scDREAMER-Sup.** scDREAMER-Sup has three additional neural networks corresponding to the Encoder $E_y$, Decoder $D_y$ and the cell type classifier $\mathcal{C}$, all with multi-layer dense neural network architecture with linear relu activation at the end of each layer. For training our model, we have adopted ADAM optimizer. The batch size of 128 was used while training the model.

**Preprocessing of scRNA-seq datasets.** The scRNA-seq datasets were preprocessed using the standard pipeline of Scanpy[44]. Raw count expression matrices were imported as Scanpy AnnData object followed by the removal of low-quality cells based on the mitochondrial gene counts. The expression matrices were then normalized (using "scanpy.pp.normalize_total" function) and log-transformed. Finally, we selected the top 2000 highly variable genes using the function "scanpy.pp.highly_variable_genes" (with flavor parameter as "seurat") as input to scDREAMER and other methods. The genes that were present across all the batches were considered. For the healthy heart atlas dataset, cells annotated as doublets were removed.

## Metrics for the evaluation of integration performance

The data integration performance of all methods was primarily evaluated based on two broad categories of metrics: (1) biological variance conservation metrics and (2) batch effect correction metrics. Following a recent study[24], we considered NMI, ARI, and ASW (cell type) as biological conservation metrics. For the evaluation of batch effect removal, we considered four metrics: ASW (batch), principal component regression (batch), graph connectivity, and kBET. For a holistic comparison of the performance across all the metrics in both of these categories, we introduced two composite scores and a combined composite score as described below. In addition, we computed isolated label F1 and isolated label silhouette scores for evaluating a method's ability to capture rare cell identities. We also computed single-cell level metrics: graph iLISI (for batch correction), graph cLISI (for bio-conservation), and proportion of true positive vs positive cells.

**Biological conservation metrics.** The computation of the biological conservation metrics requires clustering of the integrated data. We have used the Louvain clustering algorithm to compute the clusters with a resolution that maximizes the NMI value. The same clustering assignment is used for computing other metrics[24].

- **Normalized mutual information (NMI):** NMI compares the overlap between any two clustering assignments where the overlap is measured in terms of mutual information. The NMI value is scaled in the range of 0 to 1 based on the mean entropy for cluster assignments and cell-type labels.
- **Adjusted Rand Index (ARI):** Rand index compares two clustering assignments considering all pairs of points and finding agreement and disagreement between the clustering assignments[45]. The adjusted rand index (ARI) accounts for the randomly correct labels and it ranges from 0 to 1 with 0 corresponding to random clustering, and 1 representing a perfect match between the clusterings[46]. For ARI calculation, we compared the cell labels with the optimized NMI Louvain clusters.
- **Cell type average silhouette width (ASW):** Silhouette width for cell $i$ is given by:

$$sw(i) = \frac{b(i) - a(i)}{\max\{a(i), b(i)\}} \qquad (11)$$

where $a(i)$ is the average of distances of cell $i$ to the other cells in the same cluster and $b(i)$ is the average distance of cell $i$ from the cells in the other nearest cluster. Average silhouette width (ASW)

is calculated by averaging over all cells and it ranges between −1 to 1, where −1 or 0 indicates that clusters are overlapping and 1 corresponds to well-separated and dense clusters. As a bio-conservation metric, we computed cell type ASW by considering the cell types as the clusters and scaled between 0 to 1 by the transformation ASW = (ASW + 1)/2. The PCA reduced space is used for the calculation of distances for the corrected space output method. This metric is not applicable to the graph-based method BBKNN.

**Batch correction metrics.**

- **ASW batch:** ASW is calculated between the batches of each subset based on the cell type. Here the 0 value indicates that the batches are well-mixed. So we scaled the $ASW = 1 - abs(ASW)$. Now the scaled value 0 indicates that the batches are not well-mixed, 1 indicates that batches are well-mixed.
- **Principal component regression (PCR) batch:** In principal component regression (PCR) batch[47], for each principal component ($PC_k$) (calculated using PCA), the variance contributed by each batch $b$ is computed as:

$$Var(M|b) = \sum_{k=1}^{N} Var(M|PC_i) * R^2(PC_i|b) \qquad (12)$$

Where $Var(M|PC_i)$ indicates the variance explained by $PC_i$ on the data matrix $M$, and $R^2$ denotes the coefficient of determination calculated using a linear regression with $PC_i$ as the dependent and $b$ as the independent variable.

- **Graph connectivity** Graph connectivity measures whether cells from the same cell type are well connected in the kNN graph. First, we compute the kNN graph with all the cells. Then we create a subset kNN graph with the cells from a particular cell type and check the number of cells in its largest connected component.

$$GC = \frac{1}{|CT|} \sum_{c \in CT} \frac{|LCC(G(N_c, E_c)|}{|N_c|} \qquad (13)$$

Where $CT$ is the set of cell types, $|LCC()|$ is the number of cells in the largest connected component. $G(N_c, E_c)$ is the kNN subgraph for cell type $c$ with $N_c$ as the number of cells in cell type $c$ and $E_c$ denoting the edges in the kNN graph containing only the cells of type $c$.

- **k-nearest neighbor batch effect test (kBET):** k-nearest neighbor batch effect test (kBET)[47] evaluates how well the label configuration of the k-nearest neighborhoods of cells matches the global label configuration. kBET test is performed iteratively over a random subset of cells, and the value is calculated as the total rejection rates over all the iterations. As kBET works on the kNN graph, for the embedding-based output and corrected feature output methods, we have used $k = 50$ for calculating the kNN graph. kBET is applied to each batch separately to account for technical effects and changes in the cell type distribution across batches. For the kNN graphs containing disconnected components, kBET is calculated on each of the connected components. The number of nearest neighbors would differ for each cell in the graph-based method like BBKNN. So diffusion-based correction is used for graph-based output to get the same number of nearest neighbors for all the cells. kBET value is scaled between 0 to 1 so that the 0 value indicates poor batch mixing and 1 indicates perfect mixing of cells. It is important to note that for the healthy heart atlas dataset, kBET could not be computed for some of the methods due to computational limitations in handling large number of cells and batches (runs getting killed even after running for 72 h on Intel Xeon Gold 6246 CPU with 512

GB memory). This behavior of *kBET* has also been reported in ref. 24. For such cases, *kBET* has been excluded when computing the composite batch correction score.

**Isolated label metrics.** We adopted two isolated label scores from ref. 24 to evaluate how well a method can capture rare cell identities. Isolated labels denote the cell types that are present in the least number of batches. For each of the isolated labels, isolated F1 and isolated ASW are calculated. These two metrics reveal how well the isolated cell types are separated from the rest of the cell types after integration.

For the calculation of the isolated F1 score, first, we need to determine the clusters containing the particular isolated label. Lovain clustering is used for the clustering, and the resolution of the clustering algorithm is set such that the cluster has the largest number of isolated labels. The F1 score of isolated labels is then measured against the other cells in the cluster. F1 score ranges from 0 to 1 with 0 indicating that all the cells in the cluster are from cell types other than the isolated label and 1 indicating that all the cells are from isolated labels.

Isolated label ASW is calculated considering the isolated cell type as one cluster and all other cell types in another cluster and separately computing their ASW.

**Other metrics.**

- **Graph inverse Simpson's index LISI:** The inverse Simpson's index is a diversity score that calculates the number of neighbors needed to be visited before appearing in the same batch again. The value of this metric ranges from 1 to B (number of batches). This LISI score is used for the evaluation of both batch correction (iLISI) and bio-conservation (cLISI)[15]. To extend the LISI to the graph-based methods graph LISI[24] is used. In graph LISI, the distance over joint embeddings is replaced with the graph-based distance between the cells to get a large number of neighbors. Dijkstra's shortest path algorithm is used to calculate the shortest distance between the cells. In cLISI, inverse Simpson's index is calculated over the cell types, i.e., the number of neighbors one needs to visit before getting the same cell type again in the neighborhood, whereas iLISI is calculated over the batches.

- **Proportion of positive and true positive cells:** We further adopted another single-cell level summary metric from ref. 22 to evaluate the integration performance. For this, first, the cells are classified into positive and negative cells. Cells surrounded by the same cell type are classified as positive; otherwise, they are classified as negative. Considering the subset of positive cells only, a positive cell is classified as true positive if the distribution of batches around it is the same as the global batch distribution. The proportion of positive cells indicates the extent of biological conservation. In contrast, the true positive percentage indicates how well the batches are well-mixed. These two proportions together serve as quantitative metrics for evaluating the batch effect removal performance.

**Composite scores.** We have introduced two composite score metrics for a holistic comparison of the performance across all the metrics for biological conservation and batch correction respectively. Moreover, we computed a combined composite score that measures the average performance of a method in biological conservation and batch correction. Each composite score is computed by averaging the scaled values of all the metrics in that category. For a dataset, the scaled value for each individual metric is obtained through min-max normalization across all the competing methods.

The composite scores are calculated as follows:

1. **Composite bio-conservation score** = $\left(\frac{NMI' + ASW' + ARI'}{3}\right)$

2. **Composite batch correction score** = $\left(\frac{ASW(batch)' + PCR\_batch' + graph\_connectivity' + kBET'}{4}\right)$

3. **Composite isolated label score** = $\left(\frac{isolated\_f1\_score' + isolated\_ASW'}{2}\right)$

4. **Combined composite score** = $\frac{1}{2}$(Composite bio-conservation score + Composite batch correction score) For any metric $M_e, M'_e$ denotes the min-max scaled value across all the competing methods.

**Metric for evaluating the accuracy of cell label prediction.** We have adopted macro F1-score to evaluate how well a method can predict the cell type labels for the datasets with missing cell type labels. The F1-score metric measures the classification accuracy as the harmonic mean of precision and recall. In the context of cell label prediction, multiple classes are involved and F1-score is computed for every cell type annotation (class). Macro F1-score computes the overall accuracy of a multi-class classifier by aggregation of F1-scores for all the individual classes. For the prediction of cell type labels, it is computed by taking the arithmetic mean of the F1-scores for all the cell type annotations.

## Competing integration methods

We have compared our method against eleven state-of-the-art integration methods: scVI[20], Scanaroma[16], Harmony[15], Seurat[17], BBKNN[18], INSCT[48], LIGER[19], iMAP[22], scANVI[30], scGEN[31] and scDML[33]. The details about the run configuration for these methods are provided in Supplementary Table 5.

## Statistics and reproducibility

No statistical method was used to predetermine the sample size. No data were excluded from the analyses. The experiments involved running computational methods on previously published, publicly available datasets and did not require randomization. The investigators were not blinded to allocation during experiments and assessment of outcome.

## Reporting summary

Further information on research design is available in the Nature Portfolio Reporting Summary linked to this article.

## Data availability

All datasets used in this study are publicly available. The human pancreas data used in this study are available in the GEO database under accession codes "GSE81076 [https://www.ncbi.nlm.nih.gov/geo/query/acc.cgi?acc=GSE81076]", "GSE85241 [https://www.ncbi.nlm.nih.gov/geo/query/acc.cgi?acc=GSE85241]", "GSE86469 [https://www.ncbi.nlm.nih.gov/geo/query/acc.cgi?acc=GSE86469]", "GSE84133 [https://www.ncbi.nlm.nih.gov/geo/query/acc.cgi?acc=GSE84133]", "GSE81608 [https://www.ncbi.nlm.nih.gov/geo/query/acc.cgi?acc=GSE81608]", and the ArrayExpress database under the accession code "E-MTAB-5061 [https://www.ebi.ac.uk/biostudies/arrayexpress/studies/E-MTAB-5061]"[24]. The lung atlas data is available in the GEO database under accession code "GSE130148 [https://www.ncbi.nlm.nih.gov/geo/query/acc.cgi?acc=GSE130148]". The human immune data used in this study are available in the GEO database under accession codes "GSE120221 [https://www.ncbi.nlm.nih.gov/geo/query/acc.cgi?acc=GSE120221]", "GSE107727 [https://www.ncbi.nlm.nih.gov/geo/query/acc.cgi?acc=GSE107727]", "GSE115189 [https://www.ncbi.nlm.nih.gov/geo/query/acc.cgi?acc=GSE115189]", "GSE128066 [https://www.ncbi.nlm.nih.gov/geo/query/acc.cgi?acc=GSE128066]" and "GSE94820 [https://www.ncbi.nlm.nih.gov/geo/query/acc.cgi?acc=GSE94820]" and in the website of 10X Genomics (PBMC10k: https://support.10xgenomics.com/single-cell-gene-expression/datasets/3.0.0/pbmc_10k_v3)[24]. The processed human pancreas, human immune, and lung atlas datasets are available at

https://figshare.com/articles/dataset/Benchmarking_atlas-level_data_integration_in_single-cell_genomics_-_integration_task_datasets_Immune_and_pancreas_/12420968[24]. The Macaque Retina dataset is available at https://singlecell.broadinstitute.org/single_cell/study/SCP212/molecular-specification-of-retinal-cell-types-underlying-central-and-peripheral-vision-in-primates#study-download. The Healthy Heart data used in this study is available at https://www.heartcellatlas.org/[32]. The processed Healthy Heart data used in this study can be downloaded from https://figshare.com/articles/dataset/Batch_Alignment_of_single-cell_transcriptomics_data_using_Deep_Metric_Learning/20499630/2[33]. The Human and Mouse cell Atlas datasets used in the study are available at https://figshare.com/articles/dataset/MCA_DGE_Data/5435866 and https://figshare.com/articles/dataset/HCL_DGE_Data/7235471, respectively. The processed Human-Mouse cell atlas data can be downloaded from https://github.com/lkmklsmn/insct/tree/master/reproducibilty[48]. All the processed datasets for the missing label experiments (for lung atlas, human immune and heart atlas tasks) are openly available at https://doi.org/10.6084/m9.figshare.24354295. The details of the experimental biological datasets used in this study are further provided in Supplementary Table 1. All other data supporting the findings of this study are available within the article and its supplementary files. Any additional requests for information can be directed to, and will be fulfilled by, the corresponding author. Source data are provided with this paper.

## Code availability

The source code and usage tutorial for scDREAMER are freely available at https://github.com/Zafar-Lab/scDREAMER, the code has also been deposited via Zenodo (https://doi.org/10.5281/zenodo.10021620)[49]. All analysis and results presented in the manuscript are available at https://github.com/Zafar-Lab/scDREAMER-reproducibility, which have also been deposited via Zenodo (10.5281/zenodo.10021936)[50].

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

## Acknowledgements

This work was partially supported by the Science and Engineering Research Board (SERB), Government of India (SRG/2020/001333) and IIT Kanpur initiation grant (IITK /CS /2019236) to H.Z. Funding for open access charge is obtained from IIT Kanpur Research I Foundation grant 20030163.

## Author contributions

H.Z. designed the study. A.S., M.K.P., and H.Z. developed the model and algorithm. A.S. and M.K.P. implemented the software and performed all experiments. All authors wrote and approved the manuscript.

## Competing interests

The authors declare no competing interests.
