## [Peer review file · Nature Communications]

REVIEWER COMMENTS

Reviewer #1 (Remarks to the Author):

This paper presents a data integration framework scDREAMER, which employs an adversarial variational autoencoder for learning lower-dimensional cellular embeddings and a batch classifier neural network for the removal of batch effects. The method is technically sound. The result seems good. Significant details are provided. The source code is provided. However, the method is not novel. The paper cited ref. [22] (<https://genomebiology.biomedcentral.com/articles/10.1186/s13059-021-02280-8>), but did not discuss it or compare with it. In fact, Ref. 22 used a very similar method for batch correction, i.e., deep autoencoders and generative adversarial networks. The novelty for this paper is not justified. In addition, there are many batch correction methods. In terms of comparing with other methods, both the depth and breath are limited (ref. <https://genomebiology.biomedcentral.com/articles/10.1186/s13059-019-1850-9>). Hence, the overall value of this paper is limited.

Reviewer #2 (Remarks to the Author):

Summary:

The paper proposed an adversarial generative model for the integration of scRNA-seq. It achieved state-of-the-art results in challenging datasets with skewed cell types, nested batch effects, and a large number of batches. It also supports the identification of rare cell types. However, I don't think the current manuscript merits publication from Nature Comm given the following concerns:

Main concern:

1. Novelty: many similar architectures of adversarial VAE have been proposed in several existing works. None of the proposed modules in the paper is original. Therefore, the novelty of the method is limited.

To be specific, the whole idea of adversarial VAE with the same Bhattacharyya distance objective has been introduced in

- "Lin, E., Mukherjee, S. & Kannan, S. A deep adversarial variational autoencoder model for dimensionality reduction in single-cell RNA sequencing analysis. BMC Bioinformatics 21, 64 (2020)."

The only major difference to the above reference is the second discriminator of the so-called "batch classifier". However, again the same "batch classifier" has been introduced in

- "Wang, Dongfang, et al. iMAP: integration of multiple single-cell datasets by adversarial paired transfer networks. Genome Biology 22.1 (2021): 1-24."

So, in this sense, the proposed method seems to be a direct combination of the above two existing approaches and re-examined on several new datasets.

On the other hand, the author did cite the above two papers in the manuscript, in references 35 and 22. However, the author didn't clearly point out in the main text the direct connections to them among the various aspects regarding encoder, decoder, objectives, and general workflows. I only noticed the mention of the inspiration of Bhattacharyya distance from reference 35 in the method section.

In general, I think the author should carefully connect this work to other adversarial training approaches for scRNA-seq, because those are the most related ones regarding the methodology.

There are other related existing adversarial training methods, just to name one,

- "Marouf, Mohamed, et al. Realistic in silico generation and augmentation of single-cell RNA-seq data

using generative adversarial networks. Nature communications 11.1 (2020): 1-12.”

I would encourage the author to carefully discuss the connections and differences between these related works.

2. The evaluation metrics and choices of baseline methods used in the work clearly follow the benchmarking paper of “Luecken, Malte D., et al. Benchmarking atlas-level data integration in single-cell genomics. Nature methods 19.1 (2022): 41-50.”. However, since scANVI (Xu, C. et al. Probabilistic harmonization and annotation of single-cell transcriptomics data with deep generative models. Mol. Syst. Biol. 17, e9620 (2021).) has been ranked as the state-of-the-art method in the benchmarking paper, why didn't the author include the scANVI in the comparison? Additional comparison will help evaluate the relative strength of the proposed method.

3. Can the proposed model works for novel cell types not in training? The author can examine this by splitting held-out cell types into the test set.

4. Can the author conduct an ablation study of different modules, particularly, for the two discriminators? Currently, it is difficult to tell the contribution of each module in the proposed approach.

5. The proposed method consistently shows sub-optimal performance regarding several batch correction-related metrics, particularly ASW label/batch and graph connectivity. See supplementary figure 2b, supplementary figure 5b, supplementary figure 8b, and supplementary figure 10b. For example, in sup figure 2b ASW label/batch, the proposed method is ranked 5th out of all 6 methods. So, does this result indicate suboptimal batch removal capability?

6. Related to the previous question. The choice of using a simple average score for batch correction in line 515 can be questionable due to the fact that the four different batch effect removal metrics constantly show controversial scores, where one method ranked very high regarding one metric may perform otherwise regarding another metric. Sometimes, the difference is even dramatic. This phenomenon is common in supplementary figure 2b, supplementary figure 5b, supplementary figure 8b, and supplementary figure 10b. To give one example, in sup fig 8b, the Scanorama method has the highest ASW label/batch score while it shows close to zero PCR batch score. Therefore, instead of simply reporting individual or average scores, I would encourage the author to examine and explain the reliability of those metrics. It will be clearer to tell the superiority of a method by a weighted comparison of the reliable metrics that make sense in individual scenarios.

Reviewer #3 (Remarks to the Author):

scDREAMER is a deep learning model for correcting batch effects from complex, atlas-scale single-cell RNA-seq datasets. scDREAMER architecture contains a variational autoencoder which can model scRNA-seq as a ZINB or NB distribution, a discriminator for comparison of measured and reconstructed scRNA-seq data and another discriminator for classification of batches. Thus the architecture is explicitly uses adversarial components for both batch and reconstruction. The authors compare the performance of scDREAMER against widely used state-of-the-art batch correction approaches and demonstrate excellent performances, often outperforming most methods across metrics. A variety of datasets with different degrees and types of complexities are benchmarked.

There are no doubt a large number of single-cell batch correction approaches, but scDREAMER attempts to tackle particularly hard problems when batch effects are complex. With some clarifications and additions as detailed below, I believe scDREAMER will be an excellent addition to the single-cell analysis toolkit.

1. The architecture used by scDREAMER is very interesting and it will informative to explore the

contribution of different components to the performance. Are the two discriminators truly necessary and what is the performance if only one of the discriminators is used? Are there characteristics of the datasets that are better corrected by one or the other discriminators?

2. How were the parameters and architecture for the different neural networks as detailed in Supplementary Table. 2 determined? How sensitive are the results to some variations in these parameters?

3. I think the performance is benchmarked quite clearly but the biological insights that could be uniquely identified by scDREAMER is not sufficiently articulated. I suggest the authors take a deeper dive into one of the many comparisons they present to highlight some unique biology that is masked in other batch correction approaches.

4. The presentation of performance as bar plots make it very hard to compare across datasets and methods. The authors should present results in a manner similar to Luecken et. al. (Nature Methods 2022) for easier comparisons

5. The utility of computational methods depend on both technical soundness and manageable time complexity. The run-time information provided by authors is quite limited and more information should be provided for users to understand the resource requirements. Specifically, the authors should present complete run-time for different datasets and methods.

Point-by-Point Responses for Reviewers

Reviewer #1 (Remarks to the Author):

R1.C1

This paper presents a data integration framework scDREAMER, which employs an adversarial variational autoencoder for learning lower-dimensional cellular embeddings and a batch classifier neural network for the removal of batch effects. The method is technically sound. The result seems good. Significant details are provided. The source code is provided. However, the method is not novel. The paper cited ref. [22] (<https://genomebiology.biomedcentral.com/articles/10.1186/s13059-021-02280-8>), but did not discuss it or compare with it. In fact, Ref. 22 used a very similar method for batch correction, i.e., deep autoencoders and generative adversarial networks. The novelty for this paper is not justified.

Response: We thank the reviewer for the positive remark about our method. However, we respectfully disagree with the reviewer regarding the novelty of our method. Our deep generative model is the first to employ two levels of adversarial training for better representation learning of cellular embeddings and batch correction respectively. The method referred to by the reviewer is iMAP. iMAP employs one encoder and two generator networks but it does not have a batch classifier network. For iMAP, the encoder network extracts the low-dimensional representations of a cell. The generators are fed with the low-dimensional representations of the cell and a batch indicator. If the batch indicator is true, the original expression profile is reconstructed, if a random batch indicator is inputted, the generators reconstruct a fabricated expression profile. Thus, no batch classification is happening in iMAP. Instead, by virtue of the adversarial generator network, the encoder is expected to capture batch-ignorant representations of the cells.

In comparison with iMAP, scDREAMER's batch classifier is not a generator network. Our batch classifier classifies the latent cellular embeddings as learned by our adversarial VAE to a particular batch. This batch classifier network learns the batch information of the cells from the learned embedding and it is adversarially trained along with the encoder. Consequently, the loss function utilized by scDREAMER's batch classifier (cross-entropy loss) is completely different from the loss function utilized by iMAP (content loss). In summary, scDREAMER's batch classifier is a novel non-generative network that is adversarially trained to learn the batch information of a cell, whereas iMAP's generative networks learn the technical noise of each batch. To further differentiate scDREAMER's ability against that of iMAP, we compared scDREAMER's performance against that of iMAP for all the benchmarking datasets and scDREAMER outperformed iMAP by a large margin for all the datasets both in terms of biological conservation and batch correction. The comparison of iMAP against that of scDREAMER can be found in the Figures and Supplementary Figures of the revised manuscript. In the revised manuscript, we have also introduced a supervised version of scDREAMER which employs the cell type labels during data integration, and this version further improves the biological conservation metrics while retaining performance in batch correction. In summary, scDREAMER has different novel components as compared to iMAP and in terms of performance also, its superiority over iMAP can be observed across all benchmarking datasets.

R1.C2

In addition, there are many batch correction methods. In terms of comparing with other methods, both the depth and breadth are limited (ref. <https://genomebiology.biomedcentral.com/articles/10.1186/s13059-019-1850-9>). Hence, the overall value of this paper is limited.

Response: We respectfully disagree with the reviewer regarding the overall value of our paper. We agree with the reviewer that there are many batch correction methods. However, a recent comprehensive benchmarking of the integration/batch correction methods (Luecken et al. Nature Methods 2022) revealed the limitations of the current methods for removing complex, nested batch effects and emphasized the need for the development of novel methods. In the previous version of our manuscript, we did benchmark our method against 6 state-of-the-art unsupervised data integration methods that were found to be top performers across different data integration tasks in that recent benchmarking paper (Luecken et al. Nature Methods 2022). The paper referred by the reviewer is a comparatively old benchmarking of batch correction methods and did not include comparison on any of the deep learning-based methods such as scVI and INSCT. Moreover, this benchmarking paper also found LIGER, Harmony, and Seurat to be the top batch mixing methods. Out of these three methods, scDREAMER was already benchmarked against Harmony and Seurat and was demonstrated to outperform them. In the revised manuscript, we have further added LIGER in our benchmarking experiments and scDREAMER clearly outperformed LIGER across different benchmarking datasets. Moreover, we added another recent batch correction method iMAP in our benchmarking and scDREAMER outperformed iMAP as well. In the revised manuscript, we have further introduced an extended version of scDREAMER (scDREAMER-Sup) that can utilize cell type labels for semi-supervised integration. We have compared the performance of scDREAMER-sup against top-performing supervised data integration methods (that also utilize cell type labels) such as scANVI and scGEN and we showed that scDREAMER-Sup outperformed scANVI and scGEN across different integration tasks and experimental settings. In summary, we have conducted a very thorough benchmarking of our method against top performers based on two different benchmarking papers and methods that came out after the recent benchmarking paper and showed that our method performs the best across different scenario. In addition, in the revised manuscript, we also performed more analyses to show how our method is able to capture biological insights that are masked by other batch correction methods. In light of all these new findings, we feel that our paper will be very much valuable for the community and scDREAMER will be a valuable method for performing integration of multiple batches for building cell type atlases.

Reviewer #2 (Remarks to the Author):

Summary:

The paper proposed an adversarial generative model for the integration of scRNA-seq. It achieved state-of-the-art results in challenging datasets with skewed cell types, nested batch effects, and a large number of batches. It also supports the identification of rare cell types. However, I don't think the current manuscript merits publication from Nature Comm given the following concerns:

Main concern:

R2.C1

1. Novelty: many similar architectures of adversarial VAE have been proposed in several existing works. None of the proposed modules in the paper is original. Therefore, the novelty of the method is limited.

To be specific, the whole idea of adversarial VAE with the same Bhattacharyya distance objective has been introduced in

- "Lin, E., Mukherjee, S. & Kannan, S. A deep adversarial variational autoencoder model for dimensionality reduction in single-cell RNA sequencing analysis. *BMC Bioinformatics* 21, 64 (2020)."

The only major difference to the above reference is the second discriminator of the so-called "batch classifier". However, again the same "batch classifier" has been introduced in

- "Wang, Dongfang, et al. iMAP: integration of multiple single-cell datasets by adversarial paired transfer networks. *Genome Biology* 22.1 (2021): 1-24."

So, in this sense, the proposed method seems to be a direct combination of the above two existing approaches and re-examined on several new datasets.

On the other hand, the author did cite the above two papers in the manuscript, in references 35 and 22. However, the author didn't clearly point out in the main text the direct connections to them among the various aspects regarding encoder, decoder, objectives, and general workflows. I only noticed the mention of the inspiration of Bhattacharyya distance from reference 35 in the method section. In general, I think the author should carefully connect this work to other adversarial training approaches for scRNA-seq, because those are the most related ones regarding the methodology. There are other related existing adversarial training methods, just to name one, - "Marouf, Mohamed, et al. Realistic in silico generation and augmentation of single-cell RNA-seq data using generative adversarial networks. *Nature communications* 11.1 (2020): 1-12."

I would encourage the author to carefully discuss the connections and differences between these related works.

Response: We agree with the reviewer that the adversarial VAE used in scDREAMER is indeed inspired by the method DR-A (the first paper mentioned by the reviewer) and it has already been mentioned in the manuscript. However, it needs to be noted that DRA is a dimensionality reduction method and cannot perform integration of data from different batches. While scDREAMER network utilizes the batch information for training, the same is not utilized by DRA architecture. Thus, DRA cannot perform batch correction when multi-batch data is used. Moreover, scDREAMER's adversarial VAE is also different from the architecture of DRA as it involves only one discriminator in comparison to two discriminators used in DRA. We further quantitatively analyzed DRA's performance for the benchmarking datasets and found it to perform very poorly as compared to scDREAMER. In fact, for the human-mouse cross-species integration task, it could not run and produced an error. The performance comparison of DRA and scDREAMER in terms of different metrics for two datasets with simple and complex batch effects (Pancreas and Human Immune respectively) is shown below.

Figure: Comparison of bio-conservation and batch correction metrics for scDREAMER and DR-A for the Human Immune and Pancreas integration tasks.

We have not included these in the main manuscript given the fact that DRA is not an integration method and it performed poorly when applied on integration benchmarks.

We further respectfully disagree with the reviewer regarding the comment that scDREAMER's 'batch classifier' is the same as used by the method iMAP (the second paper mentioned by the reviewer). iMAP is indeed another integration method that employs one encoder and two generator networks, but it does not have a batch classifier network. For iMAP, the encoder network extracts the low-dimensional representations of a cell. The generators are fed with the low-dimensional representations of the cell and a batch indicator. If the batch indicator is true, the original expression profile is reconstructed, if a random batch indicator is inputted, the generators reconstruct a fabricated expression profile. Thus, no batch classification is happening in iMAP. Instead, by virtue of the adversarial generator network, the encoder is expected to capture batch-ignorant representations of the cells.

In comparison with iMAP, scDREAMER's batch classifier is not a generator network. Our batch classifier classifies the latent cellular embeddings as learned by our adversarial VAE to a particular batch. This batch classifier network learns the batch information of the cells from the learned embedding and it is adversarially trained along with the encoder. Consequently, the loss function utilized by scDREAMER's batch classifier (cross-entropy loss) is completely different from the loss function utilized by iMAP (content loss). In summary, scDREAMER's batch classifier is a novel non-generative network that is adversarially

trained to learn the batch information of a cell, whereas iMAP's generative networks learn the technical noise of each batch. To further differentiate scDREAMER's ability against that of iMAP, we compared scDREAMER's performance against that of iMAP for all the benchmarking datasets and scDREAMER outperformed iMAP for all the datasets both in terms of biological conservation and batch correction. The comparison of iMAP against that of scDREAMER can be found in the updated Figures and Supplementary Figures of the revised manuscript.

In the revised manuscript, we have added more discussion on the connections and differences between our work with existing adversarial training methods. We have further added a supplementary note (Supplementary Note 1) to discuss these connections and differences in more detail. We have cited the paper referred by the reviewer as well as other papers utilizing adversarial training for dimensionality reduction, batch integration, imputation and simulation of scRNA-seq datasets.

R2.C2

The evaluation metrics and choices of baseline methods used in the work clearly follow the benchmarking paper of "Luecken, Malte D., et al. Benchmarking atlas-level data integration in single-cell genomics. *Nature methods* 19.1 (2022): 41-50.". However, since scANVI (Xu, C. et al. Probabilistic harmonization and annotation of single-cell transcriptomics data with deep generative models. *Mol. Syst. Biol.* 17, e9620 (2021).) has been ranked as the state-of-the-art method in the benchmarking paper, why didn't the author include the scANVI in the comparison? Additional comparison will help evaluate the relative strength of the proposed method.

Response: scANVI is a supervised model for harmonization and annotation of single-cell transcriptomic data. This model is actually an extension of scVI that can leverage the cell type labels for all/ a subset of the cells present in the data sets to perform integration of cells from multiple batches. However, cell type annotation is not always available for each dataset and the inference of cell type labels is itself a different research question. scDREAMER was developed as an unsupervised batch integration method as these are more general and applicable also for cases where cell type labels are not available. As a consequence, earlier, we benchmarked scDREAMER against state-of-the-art unsupervised data integration methods only. Comparison of supervised and unsupervised algorithms will be unfair as supervised algorithms will utilize additional cell type labels for performing batch integration whereas such information will not be available to the unsupervised algorithms. Nonetheless, we have compared scDREAMER's performance against that of scANVI for the two most challenging integration tasks (lung and human immune) and we can see that by employing cell type labels, scANVI improves biological conservation but batch correction is poorer compared to that of scDREAMER. Moreover, to perform a fair comparison between these methods, we have developed an extension of scDREAMER which can utilize available cell type labels. We call this version - scDREAMER-supervised (scDREAMER-Sup in short) and it utilizes cell type labels (for all/ a subset of cells as per availability) to perform a semi-supervised integration of multiple batches. We compared the performance of scDREAMER-Sup against that of scANVI and scGEN, two state-of-the-art supervised data integration methods (as per Luecken et al. 2022) for lung and human immune integration tasks across different percentage of missing cell type labels. Our experiments show that scDREAMER-sup outperformed both scANVI and scGEN by a significant margin for both biological conservation and batch correction across all the datasets and experimental settings. These new results have been discussed in a new subsection titled **'scDREAMER-supervised utilizes available cell type labels for improved**

conservation of biological knowledge', Fig. 7, and Supplementary Figs 17-20. The Methods section of the revised manuscript contains the details of scDREAMER-Sup model.

R2.C3

Can the proposed model works for novel cell types not in training? The author can examine this by splitting held-out cell types into the test set.

Response: scDREAMER is an unsupervised batch-integration method that only takes batch information and does not utilize cell type labels during the training. Hence, prior knowledge of cell types is not used by scDREAMER and thus it can work for novel cell types not used in training. Also, for unsupervised batch integration, it is common to utilize all the available data for training the model. However, to further verify this following the suggestion of the reviewer, we assessed scDREAMER's performance in integrating two held-out cell types (alpha and delta cells respectively) that were not used in training for the pancreas integration task (Supplementary Fig. 4). As can be seen, scDREAMER was able to capture the held-out cell type in a separate cluster and this performance can also be observed in the quantitative assessment (Supplementary Fig. 4c). Given the unsupervised nature of scDREAMER, it is common to update the network weights as new data is encountered. We further quantified scDREAMER's performance when the network weights got updated after training with new cell type (i.e., alpha cells, and delta cells respectively) for 50 and 150 epochs. The representation of the held-out cells improved (Supplementary Fig. 4c) a little after including them in training. Both qualitative and quantitative assessment of the held-out cell type embeddings demonstrate scDREAMER's ability in capturing novel cell types.

R2.C4

Can the author conduct an ablation study of different modules, particularly, for the two discriminators? Currently, it is difficult to tell the contribution of each module in the proposed approach.

Response: In the revised manuscript, we have performed ablation studies to demonstrate the contribution of the components of scDREAMER i.e., the Discriminator and the Batch Classifier. Supplementary Table 2 shows the performance of scDREAMER in comparison to scDREAMER without Discriminator (scDREAMER-woDis) and scDREAMER without Batch Classifier (scDREAMER-woBC).

The ablation study shows that the discriminator contributes to better biological conservation and the batch classifier improves batch correction. Thus, by combining the two components, scDREAMER performs the best across all benchmarking datasets. The ablation study results have been discussed in a new subsection of the Results section.

R2.C5

The proposed method consistently shows sub-optimal performance regarding several batch correction-related metrics, particularly ASW label/batch and graph connectivity. See supplementary figure 2b, supplementary figure 5b, supplementary figure 8b, and supplementary figure 10b. For example, in sup figure 2b ASW label/batch, the proposed method is ranked 5th out of all 6 methods. So, does this result indicate suboptimal batch removal capability?

Response: The results do not indicate sub-optimal batch effect removal capability because we have considered the equal contribution of every metrics in composite batch-correction score to avoid any biases towards a specific batch correction metric. This is done through the min-max scaling of the metrics to the same range (0-1). Also, different metrics evaluate different aspects of batch effect correction, as a result a method which performs well in terms of one metric may not always perform well in terms of a different metric. This has also been observed in previous benchmarking of data integration methods (Luecken et al. 2022). A method which has a well-rounded performance in terms of all metrics will be better in terms of batch-correction. As for our proposed method scDREAMER, it shows decent performance in terms of all batch-correction metrics. It does not perform suboptimally for graph connectivity. scDREAMER is ranked best in terms of graph connectivity for Human mouse dataset, 2nd for Macaque retina and Pancreas dataset, and 3rd for Lung and Immune dataset. In terms of ASW label/batch, scDREAMER is ranked 3rd for Human mouse and Human immune dataset and 4th for the other datasets. However, in terms of ASW label/batch the difference of scDREAMER from the top method is not very large (0.05-0.08). Moreover, scDREAMER consistently performed well in terms of PCR batch and kBET and thus it achieved high composite batch-correction score on most datasets indicating its superior batch effect removal ability. In addition, we also computed the proportion of positive and true positive cells for scDREAMER and its superior performance (over 2nd and 3rd best methods based on combined composite score) based on these metrics further indicates its ability to remove batch effects.

R2.C6

Related to the previous question. The choice of using a simple average score for batch correction in line 515 can be questionable due to the fact that the four different batch effect removal metrics constantly show controversial scores, where one method ranked very high regarding one metric may perform otherwise regarding another metric. Sometimes, the difference is even dramatic. This phenomenon is common in supplementary figure 2b, supplementary figure 5b, supplementary figure 8b, and supplementary figure 10b. To give one example, in sup fig 8b, the Scanorama method has the highest ASW label/batch score while it shows close to zero PCR batch score. Therefore, instead of simply reporting individual or average scores, I would encourage the author to examine and explain the reliability of those metrics. It will be clearer to tell the superiority of a method by a weighted comparison of the reliable metrics that make sense in individual scenarios.

Response: For computing the composite batch correction score, before calculating the average of different batch correction metrics, they were scaled (using min-max scaling) to the same range (0-1) so that each batch correction metric has the same discriminative power and can equally contribute to the composite score. This also helps us avoid any bias towards a specific metric as different metrics evaluate different aspects of batch effect correction. For example, PCR batch computes the total variance explained by the batch variable, graph connectivity metric assesses whether a kNN graph constructed using the integrated data directly connects all cells with the same cell type, kBET determines if the k-nearest neighborhood of a cell has a similar distribution of batch labels to the expected global batch label distribution. Thus, a method which performs well in terms of one metric may not perform well in terms of other metrics. That is exactly what we observe for Scanorama, while it performed well in terms of ASW label/batch, it performed bad in terms of other three metrics. This was also observed in the recent benchmarking paper (Luecken et al. 2022). In comparison, a method which has a well-rounded performance across the different metrics can be

treated as having a better holistic batch correction performance. Also, our metric aggregation follows the same strategy as used in Luecken et al. 2022 and the best practices for ranking methods in machine learning benchmarks (Maier-Hein et al. 2018). Also, all the individual metrics are quite reliable as they have been utilized in multiple previous studies (Buttner et al. 2019, Nhu Tran et al. 2020) including Luecken et al. 2022. Thus, in computing the composite batch correction score, we are indeed following the state-of-the-art approach. While a weighted comparison of metrics is a nice idea, the determination of appropriate weights would itself be a different research question and require further experimentation with the metrics which is out of the scope of the current study.

Reviewer #3 (Remarks to the Author):

scDREAMER is a deep learning model for correcting batch effects from complex, atlas-scale single-cell RNA-seq datasets. scDREAMER architecture contains a variational autoencoder which can model scRNA-seq as a ZINB or NB distribution, a discriminator for comparison of measured and reconstructed scRNA-seq data and another discriminator for classification of batches. Thus the architecture is explicitly uses adversarial components for both batch and reconstruction. The authors compare the performance of scDREAMER against widely used state-of-the-art batch correction approaches and demonstrate excellent performances, often outperforming most methods across metrics. A variety of datasets with different degrees and types of complexities are benchmarked.

There are no doubt a large number of single-cell batch correction approaches, but scDREAMER attempts to tackle particularly hard problems when batch effects are complex. With some clarifications and additions as detailed below, I believe scDREAMER will be an excellent addition to the single-cell analysis toolkit.

Response: We thank the reviewer for the encouraging remark about our method and hope that in the revised manuscript we have added the necessary clarifications and additions requested by the reviewer.

R3.C1

The architecture used by scDREAMER is very interesting and it will be informative to explore the contribution of different components to the performance. Are the two discriminators truly necessary and what is the performance if only one of the discriminators are used? Are there characteristics of the datasets that are better corrected by one or the other discriminators?

Response: We thank the reviewer for the positive remark about our method. We have performed ablation studies for demonstrating the contributions of both the components i.e., Discriminator and Batch classifier. The results of the ablation study have been discussed in the Results section and Supplementary Table 2 shows the detailed comparison of scDREAMER against that of scDREAMER without Discriminator (scDREAMER-woDis) and scDREAMER without Batch Classifier (scDREAMER-woBC) across all the datasets. Our ablation study clearly shows the importance of both the components. While the discriminator helps to improve the biological conservation, the batch classifier helps remove batch effects. Thus, by

combining both the components, scDREAMER achieves superior performance in terms of both bio-conservation and batch-effect correction.

R3.C2

How were the parameters and architecture for the different neural networks as detailed in Supplementary Table. 2 determined? How sensitive are the results to some variations in these parameters?

Response: We have used standard values for different hyperparameters (e.g., batch size, number of training epochs, etc.) and architecture for our neural networks. Also, in our evaluations across different integration tasks and experimental settings, we have used the same values for hyperparameters and same architecture for our neural networks. This further indicates that scDREAMER requires a minimal hyperparameter tuning. We further performed robustness study for two learning rate parameters and showed that scDREAMER is robust to changes in learning rate parameters as well. These results are presented in Supplementary Table 4.

R3.C3

I think the performance is benchmarked quite clearly but the biological insights that could be uniquely identified by scDREAMER is not sufficiently articulated. I suggest the authors take a deeper dive into one of the many comparisons they present to highlight some unique biology that is masked in other batch correction approaches.

Response: We thank the reviewer for this excellent suggestion. We performed more in-depth analysis of the human immune integration task and showed that scDREAMER was able to capture distinct subtypes of dendritic cells (DC) as separate clusters. In fact, post-integration, it was able to even identify a novel DC subtype (DC5 in Villani et al.) which was present as a very small population. In comparison, second and third best performing methods scVI and Harmony failed to distinguish these dendritic cell subtypes. While scVI mixed some of the DC subtypes with completely other cell types (e.g., CD20+ B cells, CD14+ monocytes), Harmony mixed multiple DC subtypes together and failed to capture some other subtypes as separate clusters. These new results have been presented in Results section and Supplementary Fig. 11.

R3.C4

The presentation of performance as bar plots make it very hard to compare across datasets and methods. The authors should present results in a manner similar to Luecken et. al. (Nature Methods 2022) for easier comparisons.

Response: We thank the reviewer for this suggestion. In the revised manuscript, we have modified the bar plots such that the bars also indicate the associated values. We hope this new representation will make the comparisons easier.

R3.C5

The utility of computational methods depend on both technical soundness and manageable time complexity. The run-time information provided by authors is quite limited and more information should be provided for

users to understand the resource requirements. Specifically, the authors should present complete run-time for different datasets and methods.

Response: We have computed runtime of different methods for different datasets and these have been provided in Supplementary Fig. 16.

References

Büttner, M., Miao, Z., Wolf, F. A., Teichmann, S. A., & Theis, F. J. (2019). A test metric for assessing single-cell RNA-seq batch correction. *Nature methods*, 16(1), 43-49.

Luecken, M. D., Büttner, M., Chaichoompu, K., Danese, A., Interlandi, M., Müller, M. F., ... & Theis, F. J. (2022). Benchmarking atlas-level data integration in single-cell genomics. *Nature methods*, 19(1), 41-50.

Maier-Hein, L., Eisenmann, M., Reinke, A., Onogur, S., Stankovic, M., Scholz, P., ... & Kopp-Schneider, A. (2018). Why rankings of biomedical image analysis competitions should be interpreted with care. *Nature communications*, 9(1), 1-13.

Tran, H. T. N., Ang, K. S., Chevrier, M., Zhang, X., Lee, N. Y. S., Goh, M., & Chen, J. (2020). A benchmark of batch-effect correction methods for single-cell RNA sequencing data. *Genome biology*, 21(1), 1-32.

Reviewers' comments:

Reviewer #1 (Remarks to the Author):

The authors made a good effort in addressing the reviewers' comments. However, I still think the novelty of this paper is too limited for Nature Communications, as also pointed out by Reviewer 2. The improvements over existing methods (which are already crowded) are incremental. Hence, I won't be able to support publishing this paper in Nature Communications.

scDREAMER was published at <https://www.biorxiv.org/content/10.1101/2022.07.12.499846v1> in July. However, its tool received little attention (only 4 stars in <https://github.com/Zafar-Lab/scDREAMER> vs. many good new bioinformatics tools published last year received hundreds to thousands of stars). It doesn't appear to be an easy tool to use.

Reviewer #3 (Remarks to the Author):

The authors have satisfactorily addressed my concerns and I support the publication of scDREAMER in Nature Communications.

In addition, I was requested to assess the response to comments by another reviewer. While I cannot speak for them, in my opinion the authors have satisfactorily addressed those comments as well.

I suggest that you consider Communications Biology as a suitable venue for your work. To transfer your manuscript there, please use our <https://mts-ncomms.nature.com/cgi-bin/main.plex?el=A6S1CkHF1B5Kdbs1X4A9ftd99RXZFZ60oAoN9FJexmbGAZ> manuscript transfer portal. You will not have to re-supply manuscript metadata and files, unless you wish to make modifications, but please note that this link can only be used once and remains active until used. For more information, please see our http://www.nature.com/authors/author_resources/transfer_manuscripts.html?WT.mc_id=EMI_NPG_1511_AUTHORTRANSF&WT.ec_id=AUTHOR manuscript transfer FAQ page.

Note that any decision to opt in to In Review at the original journal is not sent to the receiving journal on transfer. You can opt in to *[In Review](https://www.nature.com/nature-portfolio/for-authors/in-review)* at receiving journals that support this service by choosing to modify your manuscript on transfer. In Review is available for primary research manuscript types only.

Point-by-Point Responses for Reviewers

Reviewer #1 (Remarks to the Author):

R1.C1

The authors made a good effort in addressing the reviewers' comments. However, I still think the novelty of this paper is too limited for Nature Communications, as also pointed out by Reviewer 2. The improvements over existing methods (which are already crowded) are incremental. Hence, I won't be able to support publishing this paper in Nature Communications.

Response: We thank the reviewer for the positive remark about the revised manuscript. However, we strongly disagree with the reviewer on the novelty and improvement aspect of scDREAMER. While there are other methods for performing integration, the inadequacy of the existing methods has been clearly pointed out in the benchmarking paper by Luecken et al. 2022 and is also evident in our benchmarking. Particularly, existing methods performed poorly in the presence of complex nested batch effects (e.g., lung and human immune integration), for the integration across a large number of batches (e.g., newly added heart atlas dataset) and atlas-level integration (e.g., cross-species integration).

In this revised manuscript, we have further introduced benchmarking using a Healthy Human Heart atlas dataset consisting of ~ 0.5 million cells distributed across 147 batches. Many of the existing methods (e.g., Seurat, LIGER, iMAP, scDML) were unable to handle such a large number of batches and failed to generate any result on this dataset. For this dataset, scDREAMER achieved 25.45% improvement over the second-best unsupervised method and scDREAMER-Sup achieved 11.54% and 44.94 - 55.62% improvement over the second-best supervised method in supervised and semi-supervised settings respectively (Figure R1g-j). We have discussed these new results in the main manuscript (please refer to Fig. 6, Supplementary Figures 14-18). These improvements are certainly not incremental given the multi-faceted challenges the dataset presented (Please refer to our new results section for more details on the different challenges with this dataset). In fact, given the generation of more such datasets by different labs as well as large consortiums (<https://www.science.org/doi/10.1126/science.abl4896>, Wesley et al. 2022, Kumar et al. 2023), it is essential to develop improved integration methods.

We have extended the benchmarking of our supervised integration method, scDREAMER-Sup (as well as other supervised integration methods) on the other datasets. Previously, these were benchmarked on only two datasets - lung and human immune integration. **Our extended benchmarking shows that scDREAMER-Sup consistently outperforms all the supervised as well as unsupervised integration methods across multiple benchmarking tasks.** In summary, for 12 different integration tasks across different benchmarking datasets, scDREAMER-Sup achieved the best combined composite score for all 12

tasks (on average, 30.86% improvement over the second-best method), achieved the best

Figure R1: scDREAMER integrates heart atlas cells from a large number (147) of batches. (a) Visualization of scDREAMER's latent space embeddings after the integration of 147 batches. (b) Visualization of scDREAMER's

latent space embeddings, cells are coloured based on the batch information (c) Visualization of scDREAMER-Sup's latent space embeddings, cells are coloured based on cell types (d) Visualization of scDREAMER-Sup's latent space embeddings, cells are coloured based on the batch information. Comparison of (e) composite bio-conservation score, (f) composite batch-correction score, and (g) combined composite score metrics between unsupervised (scVI, Harmony, Seurat, Scanorama, INSCCT, scDREAMER) and supervised (scGEN, scANVI, and scDREAMER-Sup) methods. Comparison of (h) composite bio-conservation score, (i) composite batch-correction score, and (j) combined composite score metrics between scGEN, scANVI and scDREAMER-Sup for different percentages of missing cell type labels for the heart atlas dataset. (k) Comparison of cell label prediction accuracy between scDREAMER-Sup and scANVI for different percentages of missing cell type labels for the heart atlas dataset. (l) Qualitative assessment of batch-mixing by visualization of scDREAMER's latent space embeddings, cells are coloured based on three categories - positive, negative and true positive. (m) Quantitative assessment of batch-mixing of scDREAMER against that of scVI and Harmony based on the percentage of positive vs. true positive cells. (n) Qualitative assessment of batch-mixing by visualization of scDREAMER-Sup's latent space embeddings, cells are coloured based on three categories - positive, negative and true positive. (o) Quantitative assessment of batch-mixing of scDREAMER-Sup against that of scANVI and scGEN based on the percentage of positive vs. true positive cells for the heart atlas dataset.

bio-conservation score for 11 (out of 12) tasks (on average, 21.05% improvement over the second-best method), and achieved the best batch-correction score for 10 (out of 12, our unsupervised scDREAMER model achieved the best score for the other 2 tasks) tasks (on average, 34.58% improvement over the second-best method). In terms of individual metrics too, scDREAMER-Sup performed the best for 71 (out of 84, 3 bio-conservation and 4 batch-correction metrics for each of the 12 tasks) times. In addition, scDREAMER-Sup performed the best in terms of isolated F1 score and composite isolated label score across all the benchmarking tasks for which these isolated scores could be calculated. We have updated the main manuscript, main figures and supplementary figures to discuss these new results. Please refer to Figures 5-7 and Supplementary Figures 11-23 for details. Finally, for another comparatively simpler integration task (Pancreas), for which we did not present the result of scDREAMER-Sup in our manuscript, it indeed performed the best in terms of all three composite scores (Figure R2).

Figure R2: Comparison of (a) composite bio-conservation score, (b) composite batch-correction score and (c) combined composite score metrics across different unsupervised and supervised integration methods for the integration of pancreatic islet dataset.

We have documented the percentage (%) improvement of scDREAMER and scDREAMER-Sup over the second-best competitor method for 17 different settings (includes unsupervised, semi-supervised and supervised settings) across the benchmarking datasets (see Table R1 below) and as can be seen, scDREAMER achieves 5-57% improvement over the second-best competitor method across different integration tasks. We think these improvements are not incremental but rather pointing to the efficacy and

versatility of scDREAMER in handling different integration scenarios which other methods are not capable of.

Table R1: Percentage improvement of scDREAMER and scDREAMER-Sup over the second-best competitor method across different benchmarking tasks

Sno	Dataset	Our Method	2nd best method	% improvement in combined composite score over second-best method
1	Pancreas	scDREAMER	Seurat	21.54
2	Lung	scDREAMER	scVI	5.45
3	Human Immune	scDREAMER	scVI	15.52
4	Healthy Heart	scDREAMER	scVI	25.45
5	Human Mouse	scDREAMER	scVI	26.98
6	Lung	scDREAMER-Sup	scGEN	18.34
7	Lung (10% missing cell labels)	scDREAMER-Sup	scANVI	37.01
8	Lung (20% missing cell labels)	scDREAMER-Sup	scANVI	27.46
9	Lung (50% missing cell labels)	scDREAMER-Sup	scANVI	25.10
10	Human Immune	scDREAMER-Sup	scGEN	12.47
11	Human Immune (10% missing cell labels)	scDREAMER-Sup	scGEN	36.40
12	Human Immune (20% missing cell labels)	scDREAMER-Sup	scANVI	41.05
13	Human Immune (50% missing cell labels)	scDREAMER-Sup	scANVI	30.42
14	Healthy Heart	scDREAMER-Sup	scGEN	11.54
15	Healthy Heart (20% missing cell labels)	scDREAMER-Sup	scGEN	44.94
16	Healthy Heart (50% missing cell labels)	scDREAMER-Sup	scANVI	57.62
17	Human Mouse	scDREAMER-Sup	scGEN	27.94

We further evaluated the performance of scDREAMER-Sup in predicting the cell type labels for the cells missing labels using different semi-supervised integration settings for the lung atlas (10%, 20% and 50% cell type labels missing), human immune (10%, 20% and 50% cell type labels missing) and Healthy Human Heart atlas datasets (20% and 50% cell type labels missing) and scDREAMER-Sup outperformed scANVI

(only other method that can predict missing cell type labels) for all these experimental settings (10-33% improvement over scANVI). Please refer to Figures 5h, 5p and 6k (also Fig. R1k) of the revised manuscript for more details. This comparison further shows scDREAMER-Sup’s superiority in utilizing the available cell type annotations over other supervised methods.

Moreover, to keep our benchmarking up-to-date, we have also added scDML (a newly published integration tool, published in **Nature Communications** on 21st Feb 2023, we received the review and decision of our manuscript on 27th Feb 2023) in our benchmarking and we showed that both scDREAMER and scDREAMER-Sup outperforms scDML by a large margin. In fact, scDREAMER achieved 16.67-75.76% improvement over scDML across multiple datasets and scDREAMER-Sup achieved 45-173.74% improvement over scDML across multiple datasets (Table R2). The main figures, results section and supplementary material have been updated with these new results.

Table R2: Percentage improvement of scDREAMER and scDREAMER-Sup over scDML across different benchmarking tasks

Sno	Dataset	Our Method	Combined Composite score of our method	Compared against	scDML Combined Composite score	% improvement over scDML
1	Pancreas	scDREAMER	0.79	scDML	0.63	25.40
2	Lung	scDREAMER	0.58	scDML	0.33	75.76
3	Human Immune	scDREAMER	0.67	scDML	0.39	71.79
4	Healthy Heart (140 batches)	scDREAMER	0.70	scDML	0.6	16.67
5	Human Mouse	scDREAMER	0.80	scDML	0.46	73.91
6	Lung	scDREAMER-Sup	0.90	scDML	0.33	173.74
7	Human Immune	scDREAMER-Sup	0.95	scDML	0.39	143.80
8	Healthy Heart	scDREAMER-Sup (integrated 147 batches)	0.87	scDML (integrated 140 batches)	0.6	45.00
9	Human Mouse	scDREAMER-Sup	0.87	scDML	0.46	89.13

We also argue that Reviewer 2’s comments are being misinterpreted here. Reviewer 2 requested for more clarity on the novelty of our method in the form of a comparison of scDREAMER against that of iMAP and DRA which we have already performed in our previous revision and clearly demonstrated how

scDREAMER outperforms them in all benchmarking scenarios. This is further echoed by Reviewer 3 who mentioned that we have satisfactorily addressed Reviewer 2's comments.

Given the performance improvement of scDREAMER (both unsupervised and supervised) over the existing methods and the inability of several existing methods in handling challenging datasets as demonstrated by our comprehensive benchmarking, we strongly feel that our paper will be of great interest to the readers of Nature Communications. As our framework provides a one-stop solution for different data integration tasks including unsupervised, semi-supervised and supervised, it will be widely applicable under different integration scenarios and will prove to be an important method for the analysis of cell atlases.

R1.C2

scDREAMER was published at <https://www.biorxiv.org/content/10.1101/2022.07.12.499846v1> in July. However, its tool received little attention (only 4 stars in <https://github.com/Zafar-Lab/scDREAMER> vs. many good new bioinformatics tools published last year received hundreds to thousands of stars). It doesn't appear to be an easy tool to use.

Response: We thank the reviewer for highlighting the popularity aspect of the scDREAMER's Github repository. Previously, scDREAMER was implemented using TensorFlow 1 which is no longer supported in Google Colab. Also, in our github repository, only the unsupervised version of scDREAMER was hosted and we previously did not upload the code or tutorial for scDREAMER-Sup. Furthermore, our BioRxiv submission was also not updated to reflect the revisions in our manuscript.

To address this, we have completely revamped our code and implemented scDREAMER and scDREAMER-Sup using TensorFlow 2 (TF2) library. Due to the change in implementation, we have rerun our method on all the benchmarking datasets and updated the Results section, Manuscript figures and Supplementary material accordingly. Tensorflow 2 environment will enable easy installation of our software. It will also enable other users to run our method using Google Colab. We have also introduced a readthedocs tutorial for better documentation on the software usage. Moreover, we have provided multiple Colab notebooks (runnable on a Google server) on our Github repository (<https://github.com/Zafar-Lab>) for the ease of using scDREAMER. In tutorials, we have demonstrated our software as a one-stop solution for different categories of data integration tasks including:

- a. Unsupervised integration: scDREAMER (no cell type annotations required)
- b. Supervised integration: scDREAMER-Sup (all cell type annotations available)
- c. Semi-supervised integration: scDREAMER-Sup (partially available cell type annotations)
- d. Atlas-level integration: scDREAMER (e.g., Human and Mouse atlas integration)

After taking all these measures, the popularity of our tool has already increased as can be observed in the screenshot of our Github repository below (Figure R3). In fact, it has obtained more stars as compared to some other published integration tools - scDML (12 stars, <https://github.com/eleozzi/scDML>), OCAT (17 stars, <https://github.com/bowang-lab/OCAT>).

In addition, we have also hosted all the codes used in our manuscript in a separate repository <https://github.com/Zafar-Lab/scDREAMER-reproducibility>. This repository contains different jupyter notebooks for the reproducibility of results in our manuscript.

Figure R3: Screenshot of the github repository of scDREAMER.

Having taken appropriate measures for improving the usage of our method in terms of reimplementing the code using TF2, preparation of readthedocs documentation and Colab Tutorials, we still believe GitHub's popularity is a measure of multiple factors not limited to ease of software usage, programming language, and application domain (See Hudson et al. Understanding the Factors That Impact the Popularity of GitHub Repositories, <https://ieeexplore.ieee.org/document/7816479>). It is difficult for any method to receive attention before being published. Also, there are many other factors involved that dictate how much attention a tool will receive. We strongly think that the popularity of Github repository cannot be a criterion for judging the publication of a manuscript. Many bioinformatics tools often become popular after the publication of the manuscript. We believe our Github repository will also gain more popularity after the publication of our manuscript as we are already seeing more usage of the tool after improving the documentation and updating the tutorials.

Reviewer #3 (Remarks to the Author):

The authors have satisfactorily addressed my concerns and I support the publication of scDREAMER in Nature Communications.

In addition, I was requested to assess the response to comments by another reviewer. While I cannot speak for them, in my opinion the authors have satisfactorily addressed those comments as well.

Response: We sincerely thank the reviewer for the positive remark about our efforts to address the reviewers' comments and support towards our manuscript. We also want to thank the reviewer for assessing our responses to comments by another reviewer. We appreciate you for providing constructive feedback which helped improve our manuscript and for your continued support for our manuscript.